# EDITVERSE: UNIFYING IMAGE AND VIDEO EDITING AND GENERATION WITH IN-CONTEXT LEARNING

**Xuan Ju**[1,2*] **Tianyu Wang**[1] **Yuqian Zhou**[1] **He Zhang**[1] **Qing Liu**[1] **Nanxuan Zhao**[1]
**Zhifei Zhang**[1] **Yijun Li**[1] **Yuanhao Cai**[3] **Shaoteng Liu**[1] **Daniil Pakhomov**[1] **Zhe Lin**[1]
**Soo Ye Kim**[1†] **Qiang Xu**[2,4†]

[1]Adobe Research    [2]CUHK    [3]Johns Hopkins University    [4]Shenzhen Loop Area Institute

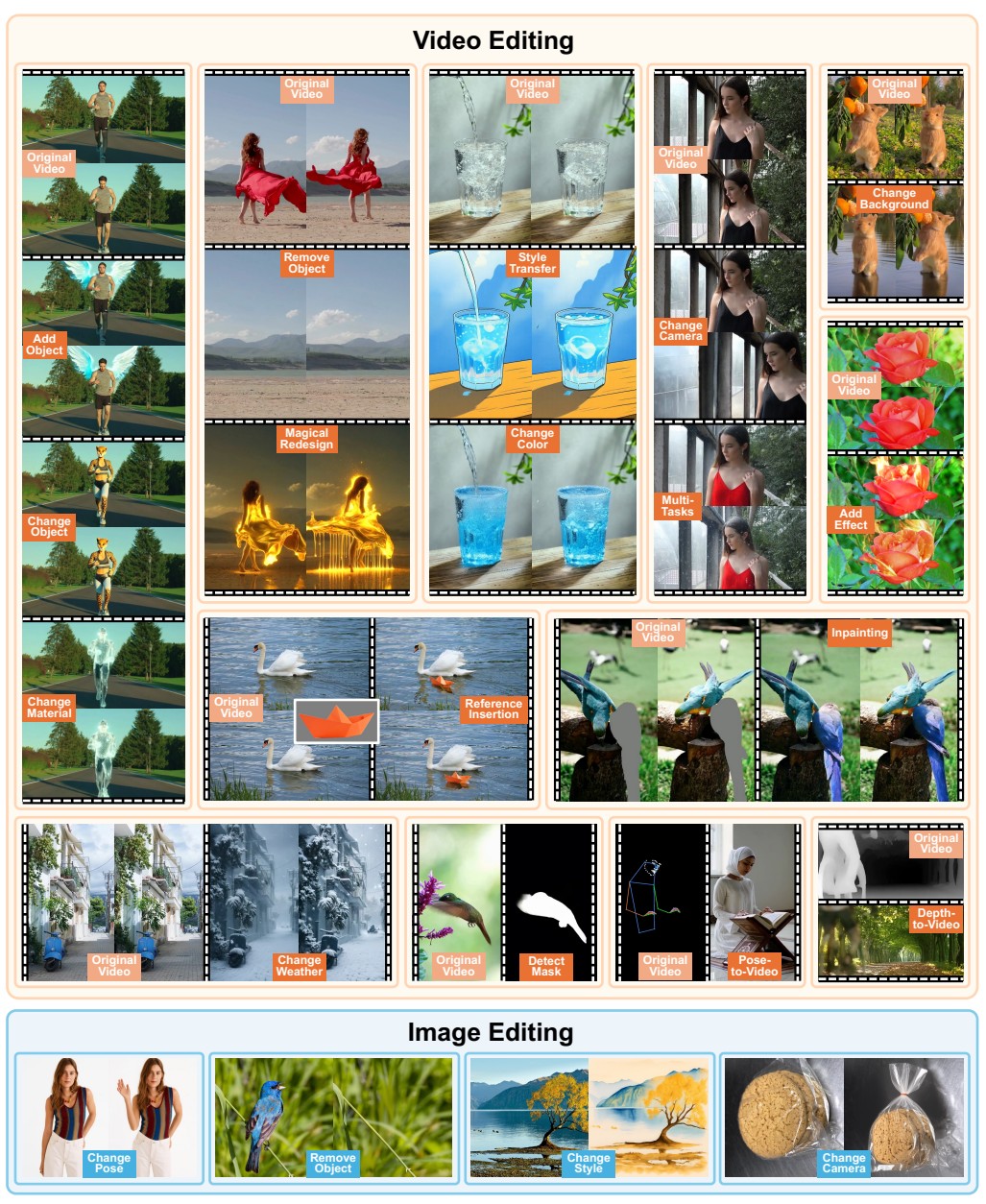

Figure 1: The strong video editing performance of EditVerse **emerges from a unified framework** trained on a diverse set of mixed image and video data. This teaser visualizes a selection of supported image and video editing tasks (Instructions in the Appendix). More results in our *Project Page*.

---

*This work was done when Xuan Ju was an intern at Adobe Research.
†Corresponding author.

## ABSTRACT

Recent advances in foundation models highlight a clear trend toward unification and scaling, showing emergent capabilities across diverse domains. While image generation and editing have rapidly transitioned from task-specific to unified frameworks, video generation and editing remain fragmented due to architectural limitations and data scarcity. In this work, we introduce EditVerse, a unified framework for image and video generation and editing within a single model. By representing all modalities, *i.e.*, text, image, and video, as a unified token sequence, EditVerse leverages self-attention to achieve robust in-context learning, natural cross-modal knowledge transfer, and flexible handling of inputs and outputs with arbitrary resolutions and durations. To address the lack of video editing training data, we design a scalable data pipeline that curates 232K video editing samples and combines them with large-scale image and video datasets for joint training. Furthermore, we present EditVerseBench, the first benchmark for instruction-based video editing covering diverse tasks and resolutions. Extensive experiments and user studies demonstrate that EditVerse achieves state-of-the-art performance, surpassing existing open-source and commercial models, while exhibiting emergent editing and generation abilities across modalities.

## 1 INTRODUCTION

Recent advancements of foundation models in computer vision and large language models highlight a clear trend toward unification and scaling (Achiam et al., 2023; Zhou et al., 2024; Deng et al., 2025), showing that joint training on diverse datasets can unlock emergent intelligence. Specifically in image generation and editing, there is also a shift from domain-specific models (Zhang et al., 2023b; Ju et al., 2023b; Li et al., 2024) toward universal models (Labs et al., 2025; Chen et al., 2025c) that unify diverse generation and editing tasks under a generalized and scalable framework.

However, unlike the image domain, the exploration of unified video generation and editing remains limited (Jiang et al., 2025; Ye et al., 2025b). This stems from two primary challenges: **(1) Architectural Limitations**: Existing video generation models, mostly based on cross-attention (Polyak et al., 2025; Wan et al., 2025) or MMDiT (Yang et al., 2024c; Kong et al., 2024) architecture, are typically designed for specific tasks such as text-to-video generation. Adapting them to support various video generation and editing tasks introduces substantial design and scaling challenges. For example, VACE (Jiang et al., 2025) uses an additional branch that accepts unedited videos and masks as input, transforming a text-to-video model into a video inpainting model. However, it relies on masks to localize the editing regions and requires task-specific input configurations, making it less practical for real-world use. To unlock emergent abilities with in-context learning, a fully unified framework must be able to process diverse input modalities (e.g., text, image, video) and types (e.g., duration, resolution) with a consistent and flexible representation. **(2) Data Scarcity and Diversity**: Unlike the vast and varied datasets readily available for image editing (Yu et al., 2024; Ye et al., 2025a; Chen et al., 2025b), high-quality and diverse video editing datasets are significantly scarce.

To address this challenge, we propose **EditVerse**, a unified framework that enables image and video editing and generation within a single model, leveraging full self-attention to enable robust in-context learning and effective knowledge transfer between images and videos. Our design considers two aspects: **(1) In-Context Learning**: We represent all modalities (text, image, and video) as a unified one-dimensional token sequence, which is then concatenated and fed into the model as a long sequence. This design enables the use of full self-attention with strong in-context learning capabilities (Ju et al., 2025) to jointly model and align different modalities. As a result, EditVerse achieves enhanced text comprehension, improved image and video editing quality, and most importantly, natural cross-modal knowledge transfer between images and videos, which effectively alleviates the limitations caused by the scarcity of video editing data. **(2) Flexibility**: We use an interleaved design for text, image, and video, inspired by the native generation architecture of multimodal large language models (MLLM), which are well-suited for supporting diverse tasks and interactive generation. This design enables the model to process image and video inputs and outputs with arbitrary resolution, temporal duration, and sequential position, thereby providing enhanced flexibility. To further distinguish positional and modal information, we introduce a four-dimensional Rotary Positional Embedding (RoPE) that incorporates sequential, temporal, height, and width dimensions.

While careful model design is crucial, simply training it on image editing data is insufficient to enable the model to perform various video editing tasks. Based on the observation that open-source instruction-based video datasets (Zi et al., 2025) are inadequate in both volume and quality, we devise a data pipeline that first generates video editing samples with task-specific models, then filters high-quality samples from the generated samples. For our unified training, we mix such curated video editing data ($232K$) with $56K$ samples filtered from Señorita-2M as well as $2M$ image generation samples, $6M$ image editing samples, and $4M$ video generation samples.

At last, due to the absence of instruction-based video editing benchmarks encompassing diverse tasks and mixed resolutions, we introduce **EditVerseBench** to enable a more comprehensive evaluation. It contains 100 videos, evenly divided between 50 horizontal and 50 vertical formats, with each video paired with two editing prompts in different editing tasks. Each data instance includes an editing instruction, a source prompt, and a target prompt, spanning 20 distinct video editing categories. Comprehensive evaluations (both automated and user studies) demonstrate that Edit-Verse achieves state-of-the-art performance compared to existing open-source methods as well as commercial models. Moreover, experiment results show the model's capacity for knowledge transfer from image to video domain and reveal emergent abilities arising from our proposed design.

## 2 RELATED WORK

**Instruction-based Image and Video Editing Datasets.** In recent years, the field has witnessed a surge in large-scale, open-source datasets for instruction-based image editing. Increasingly sophisticated data annotation pipelines have been designed and continuously improved, advancing from earlier methods doing large-scale annotation using editing models with lower success rates (*e.g.*, InstructPix2Pix (Brooks et al., 2023) and HQ-Edit (Hui et al., 2024)) or small-scale manual labeling (*e.g.*, MagicBrush (Zhang et al., 2023a)), to advanced techniques that leverage well-trained task-specific models and pipelines to generate better quality data at a large scale (*e.g.*, UltraEdit (Zhao et al., 2024), OmniEdit (Wei et al., 2024), AnyEdit (Yu et al., 2024), SEED-Data-Edit (Ge et al., 2024), and EditWorld (Yang et al., 2024b)). Data quality improvement further boosts the performance of instruction-based image editing models (OpenAI, 2024; Labs et al., 2025), which are then served as a data source of open-source datasets (*e.g.*, ShareGPT-4o-Image (Chen et al., 2025b)).

However, video editing datasets progress at a slower pace. InsV2V (Cheng et al., 2023) uses Prompt-to-Prompt (Hertz et al., 2022) and a large language model (LLM) to create its video editing datasets, where the low performance ceiling of Prompt-to-Prompt leads to poor dataset quality. Although VIVID-10M (Hu et al., 2024) provides a collection of videos with corresponding textual instructions and mask annotations, it lacks paired ground-truth edited videos, making it unsuitable for training instruction-based video editing models. Señorita-2M (Zi et al., 2025) builds an instruction-based video editing dataset using task-specific diffusion models. However, when compared to datasets in image editing, it exhibits notable limitations in both quality and editing diversity. In conclusion, datasets for instruction-based video editing are significantly less mature than the image domain, necessitating architectural innovation to transfer editing capabilities from image to video.

**Image and Video Editing.** The success of diffusion models has led to rapid progress in image and video editing. Since most pre-trained models are designed for text-to-image and text-to-video generation, early research explores training-free image and video editing techniques based on these models, often by manipulating attention maps or latent spaces (Hertz et al., 2022; Cao et al., 2023; Ju et al., 2023a; Wang et al., 2023a; Qi et al., 2023; Liu et al., 2024b; Yoon et al., 2024). Despite their simplicity, such techniques frequently yield unsatisfactory results characterized by a lack of precise control and low quality. Consequently, the field has shifted mainly to data-driven training-based methods. For image editing, methods such as InstructPix2Pix (Brooks et al., 2023) and subsequent works (Sheynin et al., 2024) concatenate the unedited image latent with the noisy latent along the channel dimension, directly fine-tuning text-to-image models for editing tasks. Later studies (Ju et al., 2025; Xiao et al., 2025; Cai et al., 2025a; Zhang et al., 2025b) find that sequential concatenation benefits model learning by using self-attention to improve in-context learning, which is a design choice that also aligns with the architectures of multimodal LLMs' native image generation (*e.g.*, BAGEL (Deng et al., 2025), transfusion (Zhou et al., 2024)). While similar techniques can be employed for video editing, investigations into instruction-based video editing are relatively rare. EVE (Singer et al., 2024a) trains adapters on top of frozen text-to-image models to enable video editing ability. InsV2V (Cheng et al., 2023) extends InstructPix2Pix (Brooks et al., 2023) to a video version. GenProp (Liu et al., 2025a) propagates the edits in the given first frame to the following

frames. Recent work UNIC (Ye et al., 2025b) concatenates conditions sequentially, similar to image editing architecture designs, and supports six editing tasks with task-aware positional embeddings. However, these methods still fall short in supporting flexible instruction-based video editing tasks.

## 3 METHOD

As illustrated in Figure 2, EditVerse employs a transformer architecture with full self-attention (Chen et al., 2025c; Ju et al., 2025). All text and vision inputs are tokenized and concatenated into a unified sequence in an interleaved manner, then fed into the model and processed with self-attention blocks (Section 3.1). To accommodate this interleaved design, we design a four-dimensional Rotary Positional Embedding with spatial (height and width), sequential, and temporal dimensions (Section 3.2). For training and inference, EditVerse predicts the visual velocity (Esser et al., 2024; Lipman et al., 2022) that guides the generation of images or videos through a denoising procedure (Section 3.3). Subsequent sections detail our framework and the insights behind our design choices.

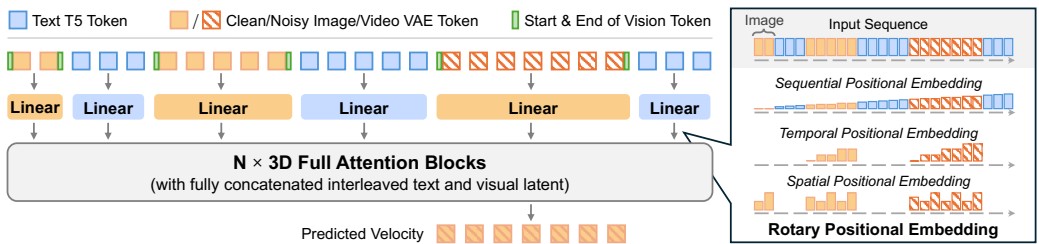

Figure 2: **Overview of EditVerse**. We design a unified framework for image and video editing and generation, which processes text and vision inputs into a unified sequence. The right part of the figure shows our positional embedding design. This framework leverages full self-attention to facilitate robust in-context learning and effective knowledge transfer among modalities.

### 3.1 INTERLEAVED TEXT AND VISION INPUT

Following prior works (Kingma & Welling, 2013), we encode the RGB pixel-space videos and images into a learned spatio-temporally compressed latent space by training a convolutional Variational Autoencoder (VAE) capable of both feature extraction and reconstruction. Specifically, given an input image or video $I_{vision}$, the VAE compresses it into a continuous-valued latent representation with downsampling ratios $r_T$, $r_H$, $r_W$. Then, the vision features are patchified into a long token sequence with a $1 \times 2 \times 2$ kernel to get $X_{vision} \in \mathbb{R}^{L_{vision} \times C_{vision}}$ ($L_{vision}$ is the vision token number, $C_{vision}$ is the channel dimension of vision feature). For a given text input $I_{text}$, we first generate text tokens using the Flan-T5-XXL (Chung et al., 2022) encoder. Then, we retain only the tokens that correspond directly to the input text, discarding the rest, yielding a final representation $X_{text} \in \mathbb{R}^{L_{text} \times C_{text}}$ ($L_{text}$ is the token count of $I_{text}$, $C_{text}$ is the channel dimension of Flan-T5-XXL), which saves computation while preserving the necessary information from text input.

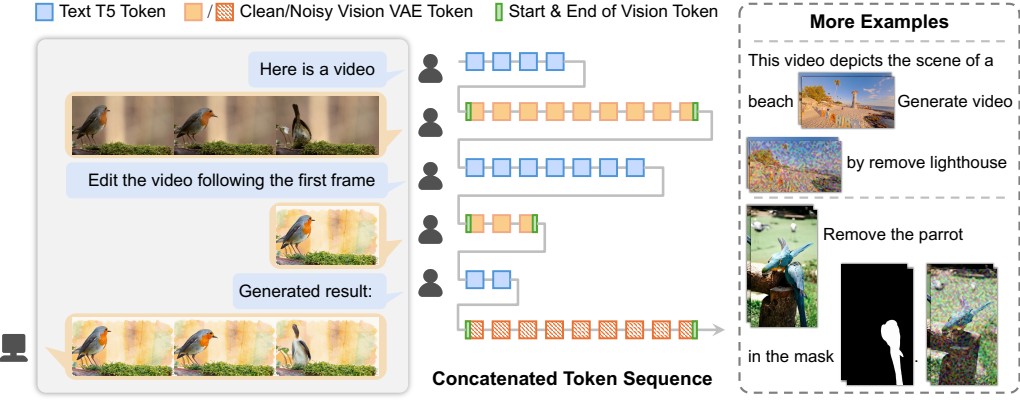

Figure 3: **Examples for the interleaved text and vision pattern**. EditVerse is capable of processing image and video inputs and outputs of arbitrary resolution, duration, and sequential positions.

To handle instructions composed of arbitrary combinations of text, images, and videos, we unify all modalities into a single interleaved sequence representation (shown in Figure 3). First, we project the tokens from each modality into a shared embedding space using separate single-layer linear projectors. This maps both text and visual inputs to the model's hidden dimension, $C$, yielding two distinct embedding matrices: $X_{text\_align} \in \mathbb{R}^{L_{text} \times C}$ and $X_{vision\_align} \in \mathbb{R}^{L_{vision} \times C}$. Then, we concatenate the projected embeddings to construct a unified input sequence, $X \in \mathbb{R}^{L \times C}$, where $L$ denotes the total number of text and vision tokens. The sequence preserves the original interleaved order of text and visual elements from the instruction. To explicitly indicate the location of vision tokens (images and videos) within an interleaved sequence, we add a learnable "start of vision" token and a learnable "end of vision" token at the beginning and the end of each vision token segment.

## 3.2 ROTARY POSITIONAL EMBEDDING

To distinguish text, image, and video from each other and to indicate their sequential positions, we design a special Rotary Positional Embedding (RoPE) that incorporates sequential, temporal and spatial (height and width) dimensions (shown in Figure 2). For each of these four positional dimensions, we apply a separate RoPE computation. (1) Sequential dimension: This dimension captures the global position within the overall sequence, starting from 0. The value is incremented by 1 for each text token and image/video frame, up to the end of the sequence. (2) Temporal dimension: This dimension is used exclusively for video frames to encode their temporal order within a video clip. It begins at 0 and increases by 1 for each subsequent frame. For text and image inputs, this dimension remains 0. (3) Height and Width Dimensions: For images and video frames, the height and width dimensions correspond to the pixel coordinates, increasing incrementally from the top-left to the bottom-right corner (Polyak et al., 2025). The increment values reflect the number of pixels along the height and width axes. For text tokens, both dimensions are set to 0. The sequential, temporal, height, and width dimensions each compute a separate RoPE, which are assigned RoPE embedding dimensions of 12, 4, 56, 56 respectively. To better support variable-length input, we use the NTK-aware interpolation (Peng et al., 2023) in RoPE calculation for context window extension.

## 3.3 TRAINING AND INFERENCE PARADIGM

Given an interleaved sequence $X_1 = \mathrm{Concat}(X_1^{(0)}, X_1^{(1)}, \ldots, X_1^{(n)})$, where each $X_1^{(i)}$ represents a clean image, a video, or a text segment, and $n$ is the total number of visual or textual segments, we randomly select one image or video $X_1^{(i)}$ as the generation target, optimizing with the Flow Matching (Lipman et al., 2022) training objective. In the diffusion process with the formulation of Flow Matching, noise sample $X_0^{(i)} \sim \mathcal{N}(0,1)$ is progressively denoised into clean data $X_1^{(i)}$ with $X_t^{(i)} = tX_1^{(i)} + (1-t)X_0^{(i)}$, where timestep $t \in [0,1]$. The learnable model $u$ is trained to predict the velocity $V_t = \frac{dX_t^{(i)}}{dt}$, which can be further derived as: $V_t = \frac{dX_t^{(i)}}{dt} = X_1^{(i)} - X_0^{(i)}$. Thus, with an input sequence $X_t = \mathrm{Concat}(X_1^{(0)}, \ldots, X_t^{(i)}, \ldots, X_1^{(n)})$, the model $u$ with parameter $\Theta$ is optimized by minimizing the mean squared error loss $\mathcal{L}$ between the ground truth velocity and the model prediction, where $X_0 = \mathrm{Concat}(X_1^{(0)}, \ldots, X_0^{(i)}, \ldots, X_1^{(n)})$:

$$\mathcal{L} = \mathbb{E}_{t, \mathbf{X}_0, \mathbf{X}_1} \left| u_\Theta(\mathbf{X}_t, t) - (\mathbf{X}_1 - \mathbf{X}_0) \right|^2$$

During inference, the diffusion model first samples $X_0^{(i)} \sim \mathcal{N}(0,1)$, then uses an ODE solver with a discrete set of $N$ timesteps to generate $X_1$ from $X_0$.

## 4 DATA PIPELINE

EditVerse is trained on large-scale data composed of: $1.9M$ image generation samples (around $2.0B$ tokens), $3.9M$ video generation samples (around $68.8B$ tokens), $6.0M$ image editing samples (around $12.6B$ tokens), and $288K$ video editing samples (around $10.2B$ tokens). Notably, the video editing datasets have significantly smaller sample number and are less diverse than image editing datasets. Thus, our architecture is specifically designed to transfer learned editing knowledge from the image domain to the video domain. We summarize the used datasets in Table 1 and provide further details below. A more detailed breakdown for each dataset is provided in Appendix Table 10.

**Video Editing Data Pipeline.** Due to the scarcity and inadequate quality of publicly available video editing datasets, we developed our pipeline to generate **EditVerse Editing Data**, which can be applied to obtain video editing pairs from any video input. (1) **Object Removal and Addition**. We first use Grounded-SAM-2 (Ravi et al., 2024; Ren et al., 2024) to extract object masks from the video. To improve the success rate of object removal, we filter candidates based on object name, total mask area, and detection confidence score. Next, we apply DiffuEraser (Li et al., 2025) to remove the masked objects. We use video pairs before and after the removal to construct object removal and addition data. (2) **Object Change**. Again, we obtain object masks from Grounded-SAM-2 (Ravi et al., 2024; Ren et al., 2024). Next, we leverage a Vision-Language Model (VLM) (Wang et al., 2024) to imagine plausible transformations of the object. We then use VACE (Jiang et al., 2025) to inpaint the masked region based on the VLM's output. To improve the success rate, we apply dynamic adjustments to the mask's shape and area, conditioned on the object's size and geometry. (3) **Style Transfer**. Previous style transfer techniques primarily rely on inference-based video editing methods (Qi et al., 2023), which we found to be unreliable when handling more diverse styles (*e.g.*, Minecraft style). To address this, we first apply an image style transfer model to edit the first frame, and then utilize VACE's (Jiang et al., 2025) depth-guided first-frame-to-video feature to generate the full styled video. (4) **Camera Change**. We select 10 camera movements and use ReCamMaster (Bai et al., 2025a) to generate camera change data. (5) **Mask Detection**. We construct the mask detection dataset by converting object removal, object addition, and object change data using the prompt template: "I want to [edit prompt]. Detect the region that needs to be edited". (6) **Propagation**. We build the propagation dataset by extracting the first edited frame from style transfer, object removal, object addition, and object change data.

**Image Datasets**

| | Dataset | #Samples |
|---|---|---|
| **Edit** | MagicBrush | $9K$ |
| | ShareGPT-4o-Image | $46K$ |
| | Object Removal & Addition[‡] | $119K$ |
| | OmniEdit | $186K(1.2M^*)$ |
| | ImgEdit | $246K(1.2M^*)$ |
| | NHR-Edit | $358K$ |
| | UltraEdit | $500K$ |
| | AnyEdit | $1.2M(2.5M^*)$ |
| | GPT-Image-Edit-1.5M | $1.5M$ |
| | Instruction-based Editing[‡] | $1.8M$ |
| **Gen** | BLIP3-o 60K | $60K$ |
| | LLaVA-pretrain | $500K$ |
| | Text-to-Image[‡] | $610K$ |
| | LLaVA-next fine-tuning | $700K$ |

**Video Datasets**

| | Dataset | #Samples |
|---|---|---|
| **Edit** | Señorita-2M | $56K(2M^*)$ |
| | EditVerse Editing Data | $232K(1.3M^*)$ |
| **Gen** | Text-to-Video[‡] | $223K$ |
| | Customization | $740K$ |
| | EditVerse Gen Data | $3.0M$ |

[*]Dataset volume before filtering. [‡] Internal dataset.

Table 1: **Statistics of the training datasets.** We mix open-source datasets, internal datasets, and EditVerse datasets for unified training. Detailed information in Table 10.

In addition, we incorporate data from the open-source dataset Señorita-2M (Zi et al., 2025). However, we observe a relatively low success rate in this dataset, necessitating extensive filtering.

**Video Generation Data Pipeline.** Since we start from a pretrained model capable of text-to-image and text-to-video tasks, we only use a small scale of pure text-based generation data ($223K$ samples for text-to-video) to preserve the model's inherent generative capabilities while simultaneously introducing controllability and enhancing its text comprehension via control tasks. For controllable video generation, we annotate control-to-video and video-to-control data pairs (including depth, sketch, and pose), where the depth map is annotated with Depth Anything v2 (Yang et al., 2024a), human pose is annotated with RTMPose (Jiang et al., 2023), and sketch is annotated with OpenCV Canny Edge Detection (Itseez, 2015). Moreover, we also include annotations for first-frame-to-video generation data and video inpainting data annotated with Grounded-SAM-2 (Ravi et al., 2024; Ren et al., 2024). The combined data from control-to-video, video-to-control, first-frame-to-video, and video inpainting are referred to as **EditVerse Gen Data** in Table 1. Additionally, we include a video customization dataset to support reference-based generation (Cai et al., 2025b).

**Image Editing.** After reviewing the data quality of existing image editing datasets, we incorporate 8 high-quality open-source datasets: MagicBrush (Zhang et al., 2023a), ShareGPT-4o-Image (Chen et al., 2025b), OmniEdit (Wei et al., 2024), ImgEdit (Ye et al., 2025a), NHR-Edit (Kuprashevich et al., 2025), UltraEdit (Zhao et al., 2024), AnyEdit (Yu et al., 2024), and GPT-Image-Edit-1.5M (Wang et al., 2025). In addition, we incorporate two internal image editing datasets: one focused on image addition and removal, and the other on free-form instruction-based image editing.

**Image Generation.** For text-to-image, we include $610K$ internal text-to-image samples as well as several open-source image understanding datasets (BLIP3-o 60K (Chen et al., 2025a), LLaVA-pretrain (Liu et al., 2023), and LLaVA-next fine-tuning (Liu et al., 2024a)) that contain high-quality text annotations, which can improve the editing instructions understanding ability.

**Data Filtering.** Since the training data is model-generated and contains errors, filtering is vital for curating high-quality examples. We used a VLM (Wang et al., 2024) to filter the dataset by scoring both editing and video quality. The scores covered instruction adherence, context preservation, video sharpness, temporal consistency, artifact presence, object integrity, aesthetics, and physical plausibility. To determine the final filtering criteria, we manually inspected the relationship between the VLM scores and the editing quality. Based on this inspection, we defined a set of score thresholds to select the final training dataset. As shown in Table 1, our video editing pipeline achieves a retention rate six times higher than Señorita-2M after filtering, demonstrating high editing quality.

## 5 EXPERIMENTS

### 5.1 IMPLEMENTATION DETAILS

EditVerse is trained on a $2B$ dense transformer architecture similar to LLaMA 3 (Dubey et al., 2024). It is initially pretrained on text-to-image and text-to-video data to get basic generative capabilities at a resolution of 360p. Then, we train the model on our dataset as listed in Section 4. For each image/video, we resize it according to its original aspect ratio so that its area falls between 256×256 and 512×512. During training, we use a global batch size of 256 and train for around $56K$ steps. We use AdamW optimizer (Loshchilov et al., 2017) with hyper-parameters set to $\beta_1 = 0.9, \beta_2 = 0.95$, a peak learning rate of $8e^{-6}$, and weight decay of $0.01$. We use a warm-up of $2K$ steps and a cosine decay learning schedule, decreasing the learning rate to the minimum of $1e^{-6}$. We set the gradient clipping norm to 1.0 and disable gradient clipping during the warm-up stage. Since the training data consist of token sequences with variable lengths, making it difficult to form batches, we adopt the packing strategy introduced in KnapFormer (Zhang et al., 2025a). During inference, we use a classifier-free guidance scale of 5.0, applying it only to text conditions. The inference timestep is set to 50 for the balance of performance and inference speed.

Figure 4: **Examples from the proposed EditVerseBench**. EditVerseBench includes 200 editing pairs, evenly distributed across 20 editing categories as well as horizontal and vertical orientations.

| Method | VLM evaluation | Video Quality | Text Alignment | | Temporal Consistency | |
|---|---|---|---|---|---|---|
| | Editing Quality ↑ | Pick Score ↑ | Frame ↑ | Video ↑ | CLIP ↑ | DINO ↑ |
| **Attention Manipulation (Training-free)** | | | | | | |
| **TokenFlow** | 5.26 | 19.73 | 25.57 | 22.70 | 98.36 | 98.09 |
| **STDF** | 4.41 | 19.45 | 25.24 | 22.26 | 96.04 | 95.22 |
| **First-Frame Propagation (w/ End-to-End Training)** | | | | | | |
| **Señorita-2M** | 6.97 | 19.71 | 26.34 | 23.24 | 98.05 | 97.99 |
| **Instruction-Guided (w/ End-to-End Training)** | | | | | | |
| **InsV2V** | 5.21 | 19.39 | 24.99 | 22.54 | 97.15 | 96.57 |
| **Lucy Edit** | 5.89 | 19.67 | 26.00 | 23.11 | 98.49 | 98.38 |
| **EditVerse (Ours)** | **7.65** | **20.07** | **26.73** | **23.93** | **98.56** | **98.42** |
| **Closed-Source Commercial Models** | | | | | | |
| Runway Aleph | 7.44 | 20.42 | 27.70 | 24.27 | 98.94 | 98.60 |

Table 2: **Quantitative comparison on EditVerseBench.** For open-source research models, we compare two training-free methods (TokenFlow and STDF), one first-frame propagation method (Señorita-2M), and one instruction-guided video editing method (InsV2V). Best results are highlighted in **bold**. We also provide the results of a commercial model, Runway Aleph. While EditVerse lags Runway Aleph in generation quality due to base model differences, our proposed method EditVerse surpasses it in editing faithfulness (via VLM evaluation on editing quality), aligning better with human judgment that is further validated by user studies shown in Figure 5.

## 5.2 EDITVERSEBENCH

Commonly used video editing benchmarks (*e.g.*, V2VBench (Sun et al., 2024), TGVE (Wu et al., 2023b; Singer et al., 2024b)) only consist of square videos and are primarily designed for training-free editing (Qu et al., 2025; Yatim et al., 2024) rather than instruction-based editing. Moreover, such benchmarks do not adequately cover the diverse editing tasks commonly encountered in real-world video editing scenarios. To address these limitations, we propose EditVerseBench, a comprehensive instruction-based video editing benchmark composed of 20 distinct instruction-based video editing tasks. We manually selected 100 videos from a free stock website (Pixabay, 2025) that cover a variety of scenes, including 50 horizontal and 50 vertical videos. For each video, we randomly select two editing instructions from the 20 editing tasks. This results in a total of 200 editing pairs (5 horizontal and 5 vertical videos per editing task). We show one example from each editing category in Figure 4. To evaluate editing performance on our proposed EditVerseBench, we use 6 metrics covering four aspects: VLM evaluation, video quality (frame-wise Pick Score (Kirstain et al., 2023)), text alignment (CLIP (Radford et al., 2021) text-image and ViCLIP (Wang et al., 2023b) text-video alignment), and temporal consistency (frame-wise CLIP (Radford et al., 2021) and DINO (Caron et al., 2021) consistency). Details can be found in the Appendix.

## 5.3 COMPARISON TO PREVIOUS METHODS

We show comparisons of EditVerseBench and TGVE+ (Singer et al., 2024b) in this section. More comparisons (*e.g.*, V2VBench (Sun et al., 2024) and image editing) are provided in the Appendix.

**Comparison on EditVerseBench.** Since InsV2V (Cheng et al., 2023) and Lucy Edit (Team, 2025) are the only open-source instruction-based video editing method that exactly matches our setting, we selected two well-known training-free methods, TokenFlow (Qu et al., 2025) and STDF (Yatim et al., 2024), as well as a first-frame propagation method, Señorita-2M (Zi et al., 2025), for comparison on EditVerseBench. We use the first frame of our results as input to Señorita-2M. Moreover, we also compare to a commercial model, Runway Aleph (Runway, 2025). As shown in Table 2, EditVerse outperforms previous research models on all metrics, demonstrating the effectiveness of our proposed method. Figure 6 shows visual comparisons on EditVerseBench. We further conduct a user study to assess human judgments of editing performance. The evaluation criteria include (i) instruction alignment,

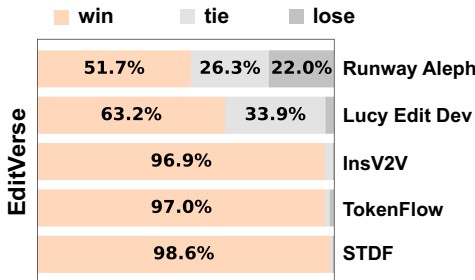

Figure 5: **User study on EditVerseBench.**

(ii) preservation of unedited regions, and (iii) overall video quality. We collected $3,000$ pairwise ratings comparing EditVerse against each of the other methods, with the results summarized in Figure 5, demonstrating the state-of-the-art performance of our proposed EditVerse. We find the user study result is more aligned with the VLM evaluation metric in automatic evaluation.

Transform the person's hair into realistic flames. The fire should flicker and flow upward, following the motion of the person's head.

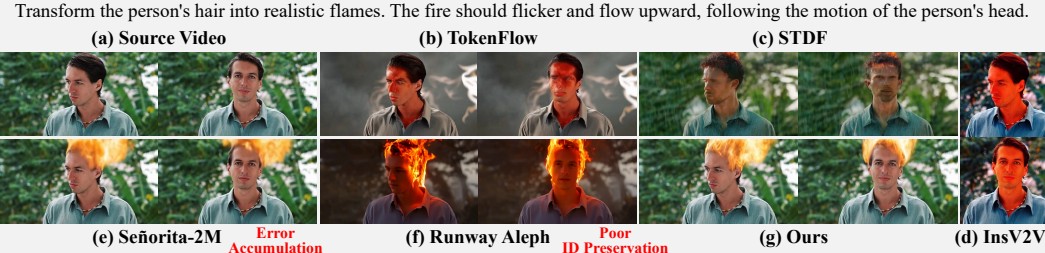

**(a) Source Video**  **(b) TokenFlow**  **(c) STDF**

**(e) Señorita-2M** Error Accumulation  **(f) Runway Aleph** Poor ID Preservation  **(g) Ours**  **(d) InsV2V**

Figure 6: **Visualization of EditVerse and other video editing methods**. EditVerse shows stronger context preservation and edit faithfulness. Complete comparisons are in the Appendix.

**Comparison on TGVE+**. Following Movie Gen (Polyak et al., 2025), we evaluate EditVerse on TGVE+ (Singer et al., 2024b). Specifically, we follow previous works and measure (i) ViCLIP$_{dir}$: text-video direction similarity, which evaluates the alignment between changes in captions and corresponding changes in the videos, and (ii) ViCLIP$_{out}$: output similarity, which measures the similarity between the edited video and the output caption. As shown in Table 3, EditVerse surpasses previous methods on both metrics. It is worth noting that all TGVE+ videos are square, whereas our training data does not include any square video editing samples.

| Method | ViCLIP$_{dir}$ ↑ | ViCLIP$_{out}$ ↑ |
|---|---|---|
| **Tune-A-Video** (Wu et al., 2023a) | 0.131 | 0.242 |
| **TokenFlow** (Qu et al., 2025) | 0.128 | 0.237 |
| **STDF** (Yatim et al., 2024) | 0.093 | 0.227 |
| **Fairy** (Wu et al., 2024) | 0.140 | 0.197 |
| **InsV2V** (Cheng et al., 2023) | 0.174 | 0.236 |
| **SDEdit** (Meng et al., 2021) | 0.131 | 0.241 |
| **EVE** (Singer et al., 2024a) | 0.198 | 0.251 |
| **Movie Gen Edit** (Polyak et al., 2025) | **0.225** | 0.248 |
| **EditVerse (Ours)** | **0.225** | **0.252** |

Table 3: **Quantitative comparison on TGVE+**. Results show superior performance of EditVerse.

### 5.4 ANALYSIS OF EMERGENT ABILITY

Emergent ability is one of the most exciting phenomena observed in large-scale model training, arising as data and model capacity increase. In this section, we specifically analyze this phenomenon.

**Demonstration of emergent ability**. We show the emergent ability of video editing in two aspects: (1) the model can perform editing tasks that were not present in the training data, and (2) for tasks included in the training data, the model's performance can even surpass the ground-truth quality.

The video editing training data covers only a limited set of tasks, including camera changes, style transfer, mask detection, object modification (addition, removal, or replacement), and propagation. However, as shown in Figure 1, our model is capable of performing tasks beyond the training distribution (*e.g.*, change material, change weather, add effects). Furthermore, it can also handle multiple tasks (*e.g.*, reference insertion by combining customization with inpainting).

We also find that EditVerse can surpass the ground-truth training data in both quality and success rate by leveraging knowledge from the image generation/editing and video generation domains. We show two examples for object removal and object change in Figure 7.

Change the squirrel to a small mouse

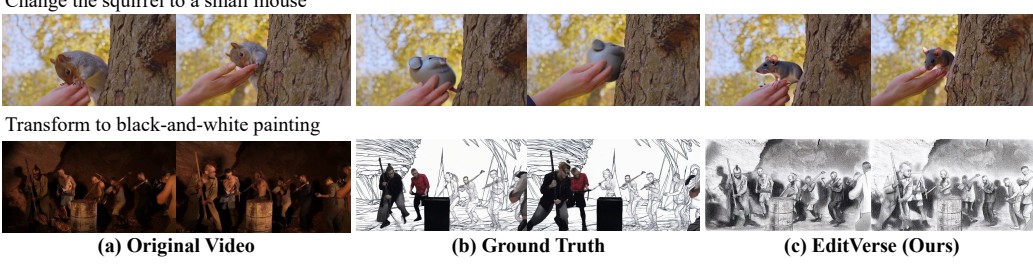

Transform to black-and-white painting

**(a) Original Video**  **(b) Ground Truth**  **(c) EditVerse (Ours)**

Figure 7: **Compare EditVerse generated results with ground truth.** Results show EditVerse can surpass ground-truth data quality by extracting knowledge from image and video generation data.

**The source of emergent ability**. We further analyze the source of emergent ability by performing ablations on the training data. We find that removing either image generation/editing data or video generation data negatively impacts video editing quality. Specifically, image generation/editing data helps the model better understand editing instructions and perform more diverse edits, while video generation data improves temporal consistency and motion modeling. Figure 8 and Table 4 illustrate the differences with and without image generation/editing and video data. Interestingly, EditVerse is able to perform some video editing tasks even without being trained on a video editing dataset.

Add a small golden crown with delicate jewels on top of the girl's head.

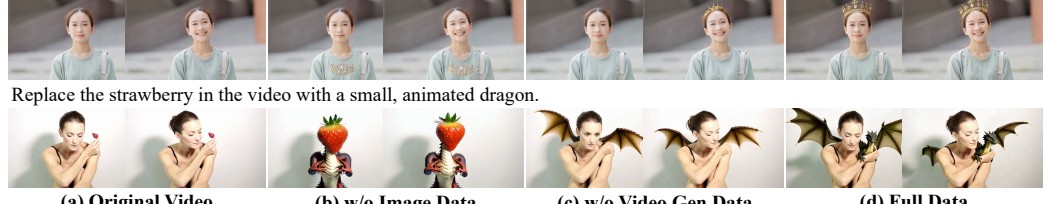

Replace the strawberry in the video with a small, animated dragon.

| (a) Original Video | (b) w/o Image Data | (c) w/o Video Gen Data | (d) Full Data |

Figure 8: **Visualization of ablation on training data.** Image data plays a critical role.

| Image | Video Gen | Video Edit | Editing Quality | Frame | Video | Pick Score | CLIP | DINO |
|---|---|---|---|---|---|---|---|---|
| ✓ | ✓ | ✗ | 3.62 | 18.64 | 22.31 | 20.44 | 93.48 | 90.27 |
| ✗ | ✗ | ✓ | 5.76 | 19.41 | 25.22 | 22.37 | 98.26 | 97.83 |
| ✓ | ✗ | ✓ | 6.52 | 19.81 | 25.78 | 22.63 | 98.24 | 97.97 |
| ✗ | ✓ | ✓ | 6.40 | 19.72 | 25.37 | 22.51 | **98.77** | **98.60** |
| ✓ | ✓ | ✓ | **6.95** | **19.99** | **26.26** | **23.81** | 98.68 | 98.44 |

Table 4: **Ablation study on training data**. We run $20K$ steps with the same setup as in Section 5.1. Results indicate that both image and video generation data are crucial to video editing performance.

## 5.5 ABLATION STUDY ON MODEL DESIGN

Compared with previous approaches (Chen et al., 2025c), our model contains two key designs: the interleaved formulation and the special positional embedding. Therefore, as shown in Table 5, we performed ablations by (i) removing the interleaved formulation (placing all images and videos at the end of the sequence) and (ii) removing the sequential dimension RoPE. Results show that both designs have a large influence on the model's performance, especially for the text alignment and editing quality. This is because the temporal consistency and video quality are partly inherited from the base model, while text alignment and editing quality largely depend on the in-context learning ability coming from the model design. Only the interleaved input format combined with sequential positional embedding can best enable the model to be aware of the relationships among different modalities (*e.g.*, knowledge transfer of image and video), thereby achieving optimal performance.

| Interleave | Sequential PE | Editing Quality | Pick Score | Frame | Video | CLIP | DINO |
|---|---|---|---|---|---|---|---|
| ✓ | ✗ | 6.42 | 19.89 | 25.77 | 22.74 | 98.62 | 98.43 |
| ✗ | ✓ | 6.84 | 19.92 | 26.19 | 23.51 | **98.69** | 98.39 |
| ✓ | ✓ | **6.95** | **19.99** | **26.26** | **23.81** | 98.68 | **98.44** |

Table 5: **Ablation study on interleaved formation and sequential RoPE**. We run $20K$ steps with the same experimental setting detailed in Section 5.1 for the ablation to save compute.

## 6 CONCLUSION

This paper introduced EditVerse, a unified framework designed to address the architectural and data-scarcity challenges in universal video generation and editing. By representing text, images, and videos as a single interleaved token sequence, our model leverages full self-attention for robust in-context learning, enabling flexible inputs/outputs of arbitrary resolution and duration, while facilitating knowledge transfer from the data-abundant image domain to the video domain.

We further developed a data pipeline for obtaining high-quality video editing samples and proposed EditVerseBench, a benchmark covering diverse editing tasks. Results show that EditVerse achieves state-of-the-art performance. These findings validate that a unified architecture can mitigate video data limitations via cross-modal learning, revealing emergent abilities and paving the way for more general multimodal foundation models. Limitations and future work are discussed in the Appendix.

**Acknowledgement.** This project was supported in part by the Innovation and Technology Fund (UD-1a-MHP/213/24), Hong Kong S.A.R.

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

# A  APPENDIX

## A.1  IMAGE AND VIDEO COPYRIGHTS

Figure 1 videos are from `pixabay` (Pixabay, 2025), stockbusters – stock.adobe.com (the first video on the top), andreybiling – stock.adobe.com (the second video on the top), and Mara Zemgaliete – stock.adobe.com (the third video on the top). Comparison images in Figure 1 are from ImgEdit-Bench (Ye et al., 2025a). Example videos in Figure 3 are from `pixabay` (Pixabay, 2025) and black-boxguild – stock.adobe.com (the first video in "More Examples"). Example videos in Figure 4, 6, and 8 are from `pixabay` (Pixabay, 2025). Adobe Stock (Adobe Inc., 2025) videos are officially licensed from the website.

## A.2  EVALUATION DETAILS

**Automatic Evaluation.** To provide a comprehensive and robust evaluation of instruction-based video editing models on EditVerseBench, we employ a suite of six metrics spanning four aspects: overall editing quality evaluated by a Vision-Language Model (VLM), video quality, text alignment, and temporal consistency.

- **Overall Editing Quality Evaluated by VLM**: To evaluate the overall editing performance, we employ a state-of-the-art Vision-Language Model (VLM), GPT-4o OpenAI (2024), to serve as an automated judge. This provides a scalable, human-like assessment that captures nuances of editing quality, such as semantic correctness and artifact presence, which are often missed by other metrics. Our method uniformly samples three frames from each source and edited video pair. For each sample, the VLM receives the source frame, the edited frame, and the text instruction. It is prompted to score the edit from 0 (worst) to 3 (best) across three key criteria: Prompt Following, Edit Quality, and Background Consistency, and then sum them together to get the overall score for this frame. The final VLM score for the entire video is the average of these three frame scores.

- **Video Quality**: We employ PickScore (Kirstain et al., 2023), which shows a strong correlation with human judgment of image quality and prompt alignment. We calculate the PickScore for each frame and average these scores across the entire video.

- **Text Alignment**: Text alignment evaluates how well the edited video reflects the given text instruction. We measure this at both the frame level and the video level.
  *CLIP Text-Image Alignment*: This metric assesses the semantic alignment between the editing instruction and each frame of the output video. We encode the text instruction using the CLIP text encoder and each frame using the CLIP vision encoder to get feature vectors, respectively. The final score is the average cosine similarity across all frames.
  *ViCLIP Text-Video Alignment*: Frame-wise alignment doesn't capture the temporal aspects of the instruction. Therefore, we use ViCLIP (Wang et al., 2023b) to compute an embedding for the entire video clip and measure its cosine similarity with the text instruction's embedding. This measures how well the video as a whole corresponds to the prompt.

- **Temporal Consistency**: Temporal consistency measures the smoothness and coherence of the edited video, penalizing flickering, jarring transitions, and inconsistent object appearances between frames. We assess this using feature similarity between adjacent frames.
  *Frame-wise CLIP Consistency*: We use the ViT-L/14 vision encoder from CLIP (Radford et al., 2021) to extract features of each frame in the edited video. The consistency score is calculated as the average cosine similarity between the features of all adjacent frames.
  *Frame-wise DINO Consistency*: To capture more fine-grained structural and textural consistency, we repeat the same procedure using features extracted from a pre-trained DINOv2 model (Caron et al., 2021). DINO's self-supervised training allows it to capture object details that might be overlooked by CLIP. Similarly, the consistency score is calculated as the average cosine similarity between the features of all adjacent frames.

**User Study.** To validate our automated metrics and directly measure human perceptual preferences, we conducted a comprehensive user study. The user study was outsourced to a professional external vendor, who recruited 20 annotators coming from diverse non-expert backgrounds. Each comparison pair was independently evaluated by 3 different annotators, and each annotator labeled 150 comparison pairs in total. To faithfully capture end-user preferences, we intentionally kept the instruction minimal and user-centric, providing only the brief prompt shown below, without additional technical guidance. This setup was chosen to reflect how typical users would judge the outputs

rather than imposing task-specific expertise. Although we did not compute a formal inter-annotator agreement statistic, the redundancy of three independent judgments per pair helps mitigate noise and increases the robustness of the aggregated preferences. We recruited detail-oriented participants to evaluate the performance of different AI video editing models. Using a web-based interface, participants were shown pairs of edited videos, labeled "Result 1" and "Result 2", each generated by different models from the same source video and text instruction. Their task was to compare the two videos and choose among "Result 1 is better," "Result 2 is better," or "They are about the same" across three evaluation criteria: (1) *Text-Instruction Alignment*: Which video better follows the provided instruction? (2) *Preservation of Unedited Regions*: Are unmodified parts of the video accurately preserved, with minimal distortion or artifacts? Ideally, edits should only affect the intended object or region. Select the one that preserves better. (3) *Aesthetic Quality*: Which video is more visually appealing in terms of realism, smoothness, and overall perceptual quality? A video is considered the winner of a comparison if it achieves a majority of wins across these three criteria. We find the user study shows a Pearson Correlation of 0.84 with automatic VLM evaluation, indicating a very strong positive correlation between the user study and VLM rankings.

**Instructions of Figure 1**. We list the editing instructions that were used in Figure 1 in a top-to-bottom, left-to-right manner: (1) Add a pair of sparkling feathered wings to the person who is running. (2) Turn the man into a running cartoon leopard. (3) Turn the person into a translucent, crystal-glass-like form. (4) Remove the woman. (5) Transform the woman's dress into a golden, fluid-like form with flames. (6) Turn into cartoon form. (7) Change the water to blue. (8) Change the camera pose to Pan Left. (9) Change the woman's slip dress to red and add a gentle snowfall effect. (10) Turn the grass into a reflective water surface. (11) Dramatically transform the scene by adding animated fiery embers and gentle flame wisps subtly dancing along the edges of the rose petals, giving the impression that the flower is being ignited by magical fire without harm, creating a surreal and striking contrast of beauty and intensity. (12) Insert a paper boat in the water [source image] A graceful white swan glides silently across the still surface of a clear lake, its long neck curved in a gentle arch and its feathers shining with a soft pearly sheen in the sunlight. Beside it, an orange paper boat drifts lightly, its sharp folds and pointed bow creating small ripples as it floats. (13) Two vibrant blue parrots are perched closely together on a tree stump. They appear to be pecking or searching for food in the crevice of the wood. The background shows a sunlit, green outdoor area with other birds visible in the distance, giving the scene a lively and natural atmosphere. (14) Change the weather to a heavy snowfall. (15) Detect the mask of the bird. (16) A young beautiful woman wearing a white hijab and a long white top sits quietly on the floor. She is reading from an open book, which rests on an intricately carved wooden stand. Her expression is calm and focused as she moves her finger along the lines of text, absorbed in her reading. The peaceful setting, with soft light and a tiled background, suggests a moment of reflection or prayer. (17) A quiet tree-lined path stretches into the distance, bathed in soft sunlight. Green leaves form a canopy overhead, while brown and yellow leaves are scattered across the ground. The scene feels calm and peaceful, inviting a slow walk or a moment of reflection in nature.

## A.3 ADDITIONAL EXPERIMENTS

**Image Editing.** We present a comprehensive evaluation of EditVerse for the task of image editing using the ImgEdit-Bench benchmark, as summarized in Table 6. The results demonstrate that EditVerse achieves highly competitive performance, surpassing a wide range of existing approaches (Deng et al., 2025; Liu et al., 2025b). This highlights the effectiveness of our method.

| Method | Add | Adjust | Extract | Replace | Remove | Background | Style | Hybrid | Action | Overall↑ |
|---|---|---|---|---|---|---|---|---|---|---|
| MagicBrush | 2.84 | 1.58 | 1.51 | 1.97 | 1.58 | 1.75 | 2.38 | 1.62 | 1.22 | 1.83 |
| Instruct-P2P | 2.45 | 1.83 | 1.44 | 2.01 | 1.50 | 1.44 | 3.55 | 1.20 | 1.46 | 1.88 |
| AnyEdit | 3.18 | 2.95 | 1.88 | 2.47 | 2.23 | 2.24 | 2.85 | 1.56 | 2.65 | 2.45 |
| UltraEdit | 3.44 | 2.81 | 2.13 | 2.96 | 1.45 | 2.83 | 3.76 | 1.91 | 2.98 | 2.70 |
| ICEdit | 3.58 | 3.39 | 1.73 | 3.15 | 2.93 | 3.08 | 3.84 | 2.04 | 3.68 | 3.05 |
| Step1X-Edit | 3.88 | 3.14 | 1.76 | 3.40 | 2.41 | 3.16 | 4.63 | 2.64 | 2.52 | 3.06 |
| UniWorld-V1 | 3.82 | 3.64 | 2.27 | 3.47 | 3.24 | 2.99 | 4.21 | 2.96 | 2.74 | 3.26 |
| BAGEL | 3.81 | 3.59 | 1.58 | 3.85 | 3.16 | 3.39 | 4.51 | 2.67 | 4.25 | 3.42 |
| **EditVerse (Ours)** | 3.81 | 3.62 | 1.44 | 3.95 | 3.14 | 3.58 | 4.71 | 2.72 | 3.80 | 3.42 |
| OmniGen2 | 3.57 | 3.06 | 1.77 | 3.74 | 3.20 | 3.57 | 4.81 | 2.52 | 4.68 | 3.44 |
| Kontext-dev | 3.83 | 3.65 | 2.27 | 4.45 | 3.17 | 3.98 | 4.55 | 3.35 | 4.29 | 3.71 |
| Ovis-U1 | 3.99 | 3.73 | 2.66 | 4.38 | 4.15 | 4.05 | 4.86 | 3.43 | 4.68 | 3.97 |
| GPT-4o-Image | 4.61 | 4.33 | 2.90 | 4.35 | 3.66 | 4.57 | 4.93 | 3.96 | 4.89 | 4.20 |

Table 6: **Quantitative comparison on ImgEdit-Bench** (Ye et al., 2025a).

**Video Generation**. We evaluate the video generation capability of EditVerse on the VBench benchmark (Zhang et al., 2024), shown in Table 7. As shown, EditVerse achieves highly competitive performance compared with a wide range of both open-source and commercial models. Notably, even though EditVerse is trained on diverse tasks beyond video generation and is built with a relatively small model size, it can still match or surpass the performance of several larger-scale systems.

| Models | # Params. | Total | Quality Score | Semantic Score |
|---|---|---|---|---|
| ModelScope | 1.7B | 75.75 | 78.05 | 66.54 |
| LaVie | 3B | 77.08 | 78.78 | 70.31 |
| OpenSoraPlan V1.3 | - | 77.23 | 80.14 | 65.62 |
| Show-1 | 6B | 78.93 | 80.42 | 72.98 |
| AnimateDiff-V2 | - | 80.27 | 82.90 | 69.75 |
| Gen-2 | - | 80.58 | 82.47 | 73.03 |
| Pika-1.0 | - | 80.69 | 82.92 | 71.77 |
| VideoCrafter-2.0 | - | 80.44 | 82.20 | 73.42 |
| EditVerse (Ours) | 2B | 80.97 | 83.47 | 70.97 |
| CogVideoX | 5B | 81.61 | 82.75 | 77.04 |
| Kling | - | 81.85 | 83.39 | 75.68 |
| Step-Video-T2V | 30B | 81.83 | 84.46 | 71.28 |
| Gen-3 | - | 82.32 | 84.11 | 75.17 |

Table 7: **Comparison with text-to-video models on the VBench** (Zhang et al., 2024). # Params. is the number of total parameters. EditVerse shows competitive performance with a small model size.

**Image Generation**. We evaluate the image generation capability of EditVerse using the GenEval benchmark (Ghosh et al., 2023) shown in Table 8, which is designed to comprehensively assess text-to-image models across multiple aspects of visual reasoning and compositional fidelity. Our method achieves state-of-the-art performance when compared against a wide range of both open-source and commercial systems, highlighting better semantically aligned generation.

| Method | Single Obj. | Two Obj. | Counting | Colors | Position | Color Attri. | Overall |
|---|---|---|---|---|---|---|---|
| **LlamaGen** | 0.71 | 0.34 | 0.21 | 0.58 | 0.07 | 0.04 | 0.32 |
| **LDM** | 0.92 | 0.29 | 0.23 | 0.70 | 0.02 | 0.05 | 0.37 |
| **SDv1.5** | 0.97 | 0.38 | 0.35 | 0.76 | 0.04 | 0.06 | 0.43 |
| **PixArt-Alpha** | 0.98 | 0.50 | 0.44 | 0.80 | 0.08 | 0.07 | 0.48 |
| **SDv2.1** | 0.98 | 0.51 | 0.44 | 0.85 | 0.07 | 0.17 | 0.50 |
| **DALL-E 2** | 0.94 | 0.66 | 0.49 | 0.77 | 0.10 | 0.19 | 0.52 |
| **Emu3-Gen** | 0.98 | 0.71 | 0.34 | 0.81 | 0.17 | 0.21 | 0.54 |
| **SDXL** | 0.98 | 0.74 | 0.39 | 0.85 | 0.15 | 0.23 | 0.55 |
| **DALL-E 3** | 0.96 | 0.87 | 0.47 | 0.83 | 0.43 | 0.45 | 0.67 |
| **Infinity**[†] | - | 0.85 | - | - | 0.49 | 0.57 | 0.73 |
| **SD3-Medium** | 0.99 | 0.94 | 0.72 | 0.89 | 0.33 | 0.60 | 0.74 |
| **FLUX.1-dev**[†] | 0.98 | 0.93 | 0.75 | 0.93 | 0.68 | 0.65 | 0.82 |
| **EditVerse (Ours)** | 0.99 | 0.95 | 0.81 | 0.82 | 0.68 | 0.64 | 0.82 |

[‡] use LLM-rewritten prompts.

Table 8: **Comparison with text-to-image models on the GenEval** (Zhang et al., 2024).

**Video Editing.** We provide a quantitative comparison on V2VBench (Sun et al., 2024) in Table 9. Noted that all V2VBench videos are square, whereas our training data does not include any square video editing samples. Our method achieves the best or competitive results across most metrics.

## A.4 DETAILED TRAINING DATA

Table 10 provides a detailed statistics overview of the whole training datasets that are used in our work, along with their respective ratio in the training process. The table is organized by task type, image editing, image generation, video editing, and video generation. For each dataset, we report the total number of samples, the ratio applied when constructing the training mixture, and a brief description highlighting the data quality, coverage, and characteristics. The training data comprises a mixture of high-quality open-source data, curated internal datasets, and filtered synthetic datasets. This combination allows us to balance scale, quality, and diversity, ultimately supporting unified training across both editing and generation tasks for images and videos.

| Method | Frames Quality ↑ | Semantic Consistency ↑ | Object Consistency ↑ | Frames Text Alignment ↑ | Frames Pick Score ↑ | Video Text Alignment ↑ | Motion Alignment ↑ |
|---|---|---|---|---|---|---|---|
| | | | Network and Training Paradigm | | | | |
| **Tune-A-Video** | **5.001** | 0.934 | 0.917 | 27.513 | 20.701 | 0.254 | -5.599 |
| **SimDA** | 4.988 | 0.940 | 0.929 | 26.773 | 20.512 | 0.248 | -4.756 |
| **VidToMe** | 4.988 | 0.949 | 0.945 | 26.813 | 20.546 | 0.240 | -3.203 |
| **VideoComposer** | 4.429 | 0.914 | 0.905 | 28.001 | 20.272 | 0.262 | -8.095 |
| **MotionDirector** | 4.984 | 0.940 | 0.951 | 27.845 | 20.923 | 0.262 | -3.088 |
| **EditVerse (Ours)** | 4.957 | **0.959** | **0.960** | **28.587** | **21.117** | **0.273** | **-3.015** |
| | | | Attention Feature Injection | | | | |
| Video-P2P | 4.907 | 0.943 | 0.926 | 23.550 | 19.751 | 0.193 | -5.974 |
| Vid2Vid-Zero | 5.103 | 0.919 | 0.912 | **28.789** | **20.950** | **0.270** | -4.175 |
| Fate-Zero | 5.036 | **0.951** | **0.952** | 25.065 | 20.707 | 0.225 | **-1.439** |
| TokenFlow | 5.068 | 0.947 | 0.943 | 27.522 | 20.757 | 0.254 | -1.572 |
| FLATTEN | 4.965 | 0.943 | 0.949 | 27.156 | 20.745 | 0.251 | -1.446 |
| FRESCO | 5.127 | 0.908 | 0.896 | 25.639 | 20.239 | 0.223 | -5.241 |
| | | | Diffusion Latent Manipulation | | | | |
| Text2Video-Zero | 5.097 | 0.899 | 0.894 | 29.124 | 20.568 | 0.265 | -17.226 |
| Pix2Video | 5.075 | 0.946 | 0.944 | 28.731 | **21.054** | **0.271** | -2.889 |
| ControlVideo | **5.404** | **0.959** | 0.948 | 28.551 | 20.961 | 0.261 | -9.396 |
| Rerender | 5.002 | 0.872 | 0.863 | 27.379 | 20.460 | 0.261 | -4.959 |
| RAVE | 5.077 | 0.926 | 0.936 | 28.190 | 20.865 | 0.255 | **-2.398** |

Table 9: **Quantitative comparison on V2VBench (Sun et al., 2024).** Methods are grouped into three categories: (i) Network and Training Paradigm, (ii) Attention Feature Injection, and (iii) Diffusion Latent Manipulation. Local best are in **bold**. Global best are underlined.

### A.5 Limitation and Future Work

While EditVerse presents a significant step toward unified video and image generation and editing, we acknowledge several limitations that open avenues for future research.

**Observed Failure Cases.** Despite its strong overall performance, EditVerse is not immune to failure cases including artifacts, flickering, low motion, logical flaws, wrong editing position, and blurred editing region. Figure 9 shows examples of two commonly seen failure types of EditVerse.

Insert an old-fashioned treasure chest half-buried among the rocks **at the man's feet**, with a faint golden glow emanating from a slightly open lid.

Remove the lamp in the background entirely from the video, …to create a natural, uncluttered look.

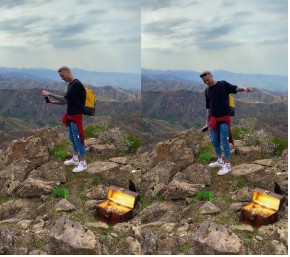
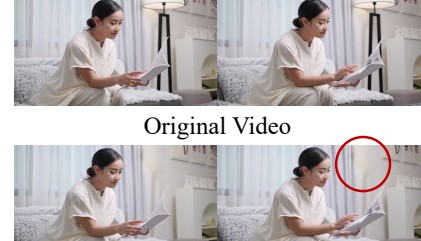

Original Video

Original Video      Edited Video      Edited Video

(a) Wrong Position      (b) Blurred Editing Area

Figure 9: **Failure case examples of EditVerse**. (a) The model fails to add object (treasure chest) at the correct position (at the man's feet). (b) Generation of blurry artifacts within the edited region.

**Computational Cost.** Our reliance on a full self-attention mechanism across a unified one-dimensional token sequence, while powerful for in-context learning, leads to significant computational overhead. The concatenation operation results in long sequence lengths, particularly for high-resolution or long-duration videos, which translates to high FLOPs and prolonged training and inference time. Future work could explore more efficient attention mechanisms to reduce the computational burden without compromising the model's cross-modal learning capabilities. Our work was not specifically designed to improve efficiency, and the reported memory usage and inference time reflect the model's raw, unoptimized performance. There are several practical ways to improve it, which we plan to explore in future work: (1) Using a higher-compression VAE. In our model,

| Dataset | #Samples | #Ratio | Information |
|---|---|---|---|
| **Image Editing** | | | |
| **MagicBrush** | 8,802 | 10 | Manually annotated with real image. High-quality. 7 editing categories. |
| **ShareGPT-4o-Image** | 46,489 | 10 | Generated by GPT-4o. 14 editing categories. Most are high-quality, but some cases contain noise. |
| **Object Removal & Addition[‡]** | 118,972 | 4 | Manually captured photos with object-present and object-absent scenes. High-quaity. |
| **OmniEdit[*]** | 185,500 | 2 | Generated by task-specific models. 7 editing categories. Good-quality but contains large noise in some editing categories. |
| **ImgEdit[*]** | 245,986 | 1 | Generated by segmentation and inpainting. 13 editing categories. Fair quality. Need filtering. |
| **NHR-Edit** | 358,463 | 5 | Generated with a designed pipeline using internal image editing model. High-quality. 17 editing categories. |
| **UltraEdit** | 500,000 | 1 | Generated by a specially designed editing model. Fair quality. 9 editing categories. |
| **AnyEdit[*]** | 1,244,033 | 1 | Generated by task-specific pipelines. 25 editing categories. Fair quality. Need filtering. |
| **GPT-Image-Edit-1.5M** | 1,500,000 | 1 | Re-process OmniEdit, UltraEdit, and HQ-Edit with GPT-4o. Most are high-quality, but some cases contain noise. |
| **Instruction-based Editing[‡]** | 1,824,969 | 1 | An internal instruction-based image editing dataset. |
| **Sum** | **6,033,214** | | |
| **Image Generation** | | | |
| **BLIP3o-60k** | 60,000 | 1 | Text-to-Image instruction tuning dataset distilled from GPT-4o. |
| **LLaVA-pretrain** | 500,000 | 1 | Text-to-Image data re-captioned using Qwen2-VL (from text-to-image-2M). |
| **Text-to-Image[‡]** | 609,950 | 1 | Internal high-quality text-to-image dataset. |
| **LLaVA-next fine-tuning** | 700,000 | 1 | Text-to-Image data generated by Flux-dev (from text-to-image-2M). |
| **Sum** | **1,869,950** | | |
| **Video Editing** | | | |
| **Camera Change** | 8,000 | 20 | Camera change data pair generated with ReCamMaster |
| **Style Transfer** | 10,327 | 10 | Style transfer data pair generated with Step1X-Edit and VACE. |
| **Mask Detection** | 15,741 | 5 | Editing region detection with prompt "I want to [edit prompt]. Detect the region that needs to be edited". Contain object removal, object addition, and object replacement. |
| **Object Chnage** | 31,482 | 10 | Object replacement data pair generated with VACE. Contain w/ mask version and w/o mask version in training. |
| **CG Removal & Addition[‡]** | 38,900 | 2 | Rendered videos with object-present and object-absent scenes. |
| **Señorita-2M[*]** | 55,711 | 2 | Generated wih task-specific models. 5 editing categories. Low quality. Need filtering. |
| **Propagation** | 59,826 | 10 | Containing editing propagation for object removal, object addition, object replacement, and style transfer. |
| **Object Removal & Addition** | 67,516 | 10 | Object removal and addition pairs generated with DiffuEraser. Contain w/ mask version and w/o mask version in training. |
| **Sum** | **287,503** | | |
| **Video Generation** | | | |
| **Depth-to-Video** | 182,097 | 2 | Depth-to-video dataset. Depth is detected with Depth Anything v2. |
| **Video-to-Depth** | 182,097 | 2 | Video-to-depth dataset. Depth is detected with Depth Anything v2. |
| **Sketch-to-Video** | 207,749 | 2 | Sketch-to-video dataset. Sketch is detected with OpenCV Canny. |
| **Video-to-Sketch** | 207,749 | 2 | Video-to-sketch dataset. Sketch is detected with OpenCV Canny. |
| **First Frame-to-Video** | 217,038 | 5 | First frame-to-video dataset. |
| **Pose-to-Video** | 233,068 | 2 | Pose-to-video dataset. Pose is detected with RTM-Pose. |
| **Video-to-Pose** | 233,068 | 2 | Video-to-pose dataset. Pose is detected with RTM-Pose. |
| **Text-to-Video[‡]** | 223,494 | 10 | Internal high-quality text-to-video dataset. |
| **Customization** | 740,111 | 1 | High-quality identity-to-video dataset from OmniVCus (Cai et al., 2025b). |
| **Video Inpainting** | 1,495,020 | 2 | Video inpainting data pair generated with Grounded SAM 2. Contain w/ mask version and w/o mask version in training. |
| **Sum** | **3,921,491** | | |

[‡] Internal datasets.
[*] We filter these datasets to improve their quality.

Table 10: **Detailed Statistics of the training datasets.** We combine high-quality open-source datasets, internal datasets, and EditVerse datasets for unified training. This table presents the dataset name, sample counts, training ratios, and key details for each dataset.

we use a VAE with relatively low compression ($8\times$ spatial downsampling), which leads to a large number of visual tokens. Recent VAEs can achieve $16\times$ spatial compression. If we replace our VAE with $16\times$ spatial compression rate models, the token length can be reduced to about one quarter, which directly lowers the attention cost. (2) Dynamic token selection. We can introduce a dynamic token selection mechanism that adaptively keeps only the most important context tokens and prunes redundant ones. This can reduce the effective sequence length for full attention, as explored in recent work FullDiT2 (He et al., 2025). (3) Distillation for faster inference. We can further apply model distillation and step distillation to reduce both the number of diffusion steps and the model size, which can noticeably speed up generation. (4) Future work could explore more efficient attention mechanisms (*e.g.*, linear attention, Mamba attention) to reduce the computational burden without compromising the model's cross-modal learning capabilities.

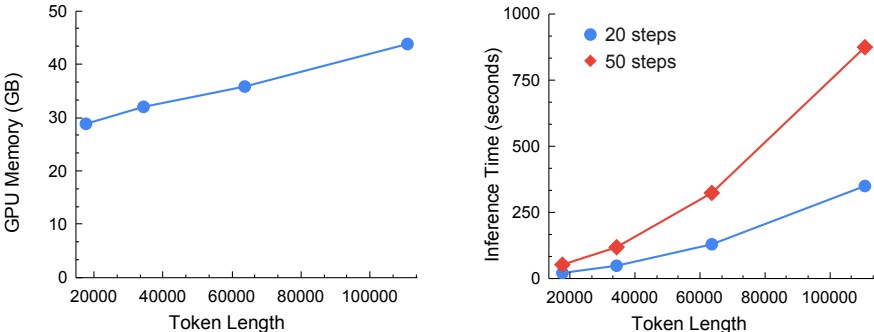

Figure 10: GPU memory usage and inference latency as functions of token length.

| Token Length | 17652 | 34342 | 63662 | 110822 |
|---|---|---|---|---|
| **Inference Time (second/20 steps)** | 21 | 48 | 129 | 349 |
| **Inference Time (second/50 steps)** | 52 | 118 | 323 | 873 |
| **GPU Memory (MB)** | 29511 | 32768 | 36663 | 44824 |
| **TFLOPs/second** | 67.9 | 58.2 | 39.4 | 25.4 |

Table 11: Inference efficiency across different token lengths

To assess the efficiency of our model with extended token length (as shown in Figure 10 and Table 11), we analyze the impact of token length on both efficiency and GPU memory usage. Experiment results demonstrate that resource consumption growth remains within a manageable and predictable range. Specifically, GPU Memory usage exhibits a strong linear relationship with token length. As the sequence length was scaled from 17,652 up to 110,822 tokens, the peak memory footprint increased modestly from 29.5 GB to 44.8 GB. This predictable and relatively slow scaling rate confirms that the extended context itself does not impose an unconstrained memory ceiling. Similarly, inference time shows a systematic, controllable increase with token length.

**Image Editing Performance.** While our unified model demonstrates strong generalization and performs on par with many image editing models, it does not currently achieve state-of-the-art performance in the image domain. Targeted optimizations, such as employing a more sophisticated data-mixing strategy or fine-tuning the model on high-quality, image-only editing datasets, could be explored to boost its performance and close the gap with specialized, state-of-the-art image editors.

**Dataset Quality.** Although our data curation pipeline is crucial for enabling instruction-based video editing, the resulting dataset contains inherent noise. The editing instructions are often concise (averaging around 10 words) and may lack the detail required for highly complex or nuanced edits. Furthermore, the automated methods used to generate editing pairs have an estimated success rate of around 65%, inevitably introducing imperfect or failed edits into the training corpus. Future efforts could focus on developing more advanced data generation and filtering techniques as mentioned in concurrent works (Wei et al., 2025; Mou et al., 2025; Bai et al., 2025b; Yang et al., 2025).

**Generalist vs. Specialist Models.** Our work highlights the potential of unified models, but it is plausible that for specific, well-defined tasks with abundant high-quality data (e.g., inpainting), a dedicated specialist model might still yield superior results.

