# OpenReview forum: "EditVerse: Unifying Image and Video Editing and Generation with In-Context Learning"
_ICLR.cc/2026/Conference — ICLR 2026 Oral_

### Official Review · Reviewer_72vX · 2025-10-22

**Soundness:** 4
**Presentation:** 4
**Contribution:** 4
**Rating:** 8
**Confidence:** 5

**Summary:**

This paper presents EditVerse, a unified framework for image and video generation and editing. By representing images, videos, and text as one-dimensional sequences of interleaved tokens, EditVerse demonstrates promising capabilities in in-context learning and cross-modal knowledge transfer. EditVerse achieves state-of-the-art results on both the EditVerseBench and TGVE+ benchmarks. Furthermore, it exhibits impressive emergent abilities when trained at scale with unified multi-modal data. Although interactive editing and generation scenarios were not evaluated in this work, I believe this approach, with appropriate scaling, could demonstrate significant potential and enable substantially broader applications. In summary, this is a technically solid and well-presented paper.

**Strengths:**

1. **Unified Design**.
The paper presents a clean solution : treat text, images, and videos as interleaved token sequences with a 4D positional embedding (sequential, temporal, height, width). This simple yet effective design handles arbitrary resolutions and durations naturally, which is quite elegant.
2. **Knowledge Transfer between Different Modality**.
The key insight is using abundant image editing data to help with scarce video editing data. Through full self-attention and mixed training (6M image editing + 288K video editing samples), the model learns to transfer knowledge across modalities.
3. **Solid Experimental Results**.
The paper presents comprehensive experiments across image/video generation and editing tasks, demonstrating that EditVerse achieves state-of-the-art performance with only 2B parameters. The ablations also clearly show both image and video data matter—images help with instruction understanding, videos help with temporal consistency. What's interesting is the emergent behavior: the model can handle tasks it wasn't explicitly trained on, and sometimes even beats the ground truth by combining knowledge from different domains.
4. **Strong Results  with Potential**.
With just 2B parameters, EditVerse achieves SOTA on EditVerseBench and TGVE+, matching or beating much larger models and commercial systems. The results hold up in both automatic metrics and user studies across 20 different editing tasks. I believe this approach has significant room to grow—with more data and parameters, it could unlock even broader applications.

**Weaknesses:**

1. **Background Drift in Local Edits**.
I noticed some background preservation issues in local editing tasks—there seems to be temporal drift in unedited regions. Could you elaborate on when and why this happens? Understanding these failure cases would be helpful.

2. **Why Text-Only Localization Work So Well?**
Most video editing models need masks for precise object editing, but EditVerse seems to work with just text instructions. This is quite impressive but also surprising. What makes this possible? Is it the scale of training, the unified architecture, or the cross-modal learning? A deeper analysis here would really strengthen the paper's contribution.

**Questions:**

See weaknesses.

---

> ### Author Response · Authors · 2025-11-21
> **Response to Reviewer 72vX**
>
> We sincerely thank the reviewer for the thoughtful and constructive feedback, as well as for the positive assessment of our unified design, presentation, knowledge transfer between different modality, solid experimental results, and strong results with potential. Below, we address each concern in detail and clarify the changes we made in the revised version (marked in red).
>
> > **Weakness 1: Background Drift in Local Edits**
> >
>
> We appreciate the reviewer’s careful inspection of our results. Our analysis shows that the primary cause of the background drift is the imperfect reconstruction quality of the underlying VAE. From our qualitative analysis, background drift is most likely to appear in the following scenarios: (1) Clips with significant camera/motion dynamics. (2) Highly cluttered or fine-textured backgrounds. In typical scenarios, these artifacts are moderate and do not change the semantics of the video, but we agree they are important to document as limitations. Even without any edit, the encode–decode process produces slight discrepancies between the input and reconstructed frames due to the VAE’s limited reconstruction fidelity. These reconstruction errors are chunk-wise and not perfectly temporally correlated, so small spatial differences become temporal flicker when viewed as a video. In future works, we will work on improving this issue by training VAE with better reconstruction quality.
>
> > **Weakness 2: Why Text-Only Localization Work So Well**
> >
>
> We thank the reviewer for highlighting this behavior. Our analysis suggests that the strong text-only localization arises from the combination of the unified architecture and the multi-modal training strategy.
>
> **(1) Unified interleaved token formulation with 4D positional encoding**
>
> EditVerse represents text, images, and videos as a single interleaved token sequence with 4D positional embeddings. Text tokens and visual tokens attend to each other in the same self-attention transformer blocks, enabling phrases like “the person on the left” or “the red car in the background” to modulate the corresponding spatial–temporal tokens directly.
>
> **(2) Knowledge transferred from rich localized image editing supervision**
>
> Our training data contains a substantial portion of localized image editing tasks. Since our model has a good knowledge transfer ability from the image domain to the video domain, this localization ability also transfers from image editing to video editing. Our ablations show that removing image editing data significantly hurts instruction alignment and localization quality in video editing, highlighting that image data is a crucial source of supervision for text-only localization.
>
> However, we also acknowledge that our model still has failure cases. For example, as shown in Fig. 9(a), the model fails to place the added object (treasure chest) at the correct location. In future work, we will mitigate such issues by training on higher-quality data and scaling the model capacity.

---

> > ### Comment · Reviewer_72vX · 2025-11-25
> > **Response to Authors**
> >
> > Thank the authors for the rebuttal, I will keep my initial positive score.

---

### Official Review · Reviewer_3VqN · 2025-10-28

**Soundness:** 3
**Presentation:** 4
**Contribution:** 3
**Rating:** 4
**Confidence:** 4

**Summary:**

This paper introduces EditVerse, a novel unified framework designed to perform both image and video generation and editing within a single model. The authors identify two primary bottlenecks in video editing: (1) architectural limitations of existing models, which are often task-specific, and (2) the scarcity of high-quality, instruction-based video editing data.

**Strengths:**

1. The core design of an interleaved 1D token sequence for mixed image/video data, combined with a full self-attention mechanism and the novel 4D RoPE, is a powerful and original solution for multimodal in-context learning.
2. The paper's standout strength is its clear demonstration that a unified model can learn video editing from data-abundant *image* editing datasets. The ablation in Table 4 (showing the model *can* edit video with zero video-edit data, albeit poorly) and the major quality drop from removing image-edit data (Fig 8) provide powerful evidence for this hypothesis.
3. The authors address the ecosystem problem by not only building a model but also creating the data (232K video-edit samples) and the evaluation tools (EditVerseBench) necessary to make progress, and they are releasing the benchmark.

**Weaknesses:**

1. The most significant weakness, which the authors acknowledge in the appendix, is the computational cost. Using full self-attention on a 1D-tokenized video (where the sequence length $L$ includes all frames) leads to $O(L^2)$ complexity. The reported 118 seconds for a single 360p video on an A100 is very slow and will likely scale quadratically or worse with resolution and duration, making it impractical for real-world use on high-resolution or long-form videos.
2. The video editing data is synthetically generated by a pipeline of specialist "teacher" models (VACE, DiffuEraser, etc.). While this is a clever solution, it means the model's performance may be capped by the quality and biases of these teachers. It's unclear if the model can perform edits that its teacher models are incapable of.

**Questions:**

1. Could the authors provide more details on the scaling properties of EditVerse? How do inference time and VRAM usage scale with (a) video duration (e.g., 3s vs. 10s vs. 30s) and (b) resolution (360p vs. 720p)? Is the full self-attention approach feasible beyond the short, low-resolution clips shown?
2. The dimension allocations for the 4D RoPE (56H, 56W, 12Seq, 4Temp) are specific. Could the authors provide a brief rationale or ablation for this design choice? For example, why is the sequential dimension (12) given more embedding space than the temporal dimension (4)?
3. How does the model perform on editing tasks or concepts that are *not* present in its synthetic "teacher" models? For instance, if VACE is poor at "making an object transparent," can EditVerse still learn this concept purely from the image-editing data and apply it successfully to video, or is its video performance on this task limited by VACE?
4. The "wrong position" failure case in Fig 9a suggests limitations in complex spatial reasoning. The VLM evaluation (Table 2) is high, but this metric averages 3 frames. How well does this VLM metric capture these kinds of high-level logical or instruction-misalignment failures, as opposed to per-frame artifacts or quality?

---

> ### Author Response · Authors · 2025-11-21
> **Response to Reviewer 3VqN - 1**
>
> We sincerely thank the reviewer for highlighting the strengths of our work, including the interleaved token and 4D RoPE design for multimodal in-context learning, the unified image-to-video editing capability enabled by unified training, and our ecosystem contributions dataset and benchmark. Below, we address each concern in detail and clarify the changes we made in the revised version (marked in red).
>
> > **Question 1 & Weakness 1: Computational cost**
> >
>
> We agree with the reviewer that efficiency and scalability are important issues. However, we believe this is not a fundamental problem of the unified formulation, and there are several practical ways to improve it, which we plan to explore in future work:
>
> - **Using a higher-compression VAE.** In our model, we use a VAE with relatively low compression (8× spatial downsampling), which leads to a large number of visual tokens. Recent VAEs can achieve 16× spatial compression. If we replace our VAE with 16× spatial compression rate models, the token length can be reduced to about one quarter, which directly lowers the attention cost.
> - **Dynamic token selection.** We can introduce a dynamic token selection mechanism that adaptively keeps only the most important context tokens and prunes redundant ones. This can reduce the effective sequence length for full attention, as explored in recent work FullDiT2.
> - **Distillation for faster inference.** We can further apply model distillation and step distillation to reduce both the number of diffusion steps and the model size, which can noticeably speed up generation.
>
> We have expanded the discussion of these directions in the revised version (Section A.4) to clarify that there are clear paths to make EditVerse more practical in real-world settings.
>
> We also incorporate a more thorough examination of efficiency, along with relevant discussions and results(we also add it in the revised version of Section A.4, marked in red). Due to the video duration and resolution can all be matched to the token length, we summarize the analysis of memory and inference time growth with token length here:
>
> | Token Length | 17652 | 34342 | 63662 | 110822 |
> | --- | --- | --- | --- | --- |
> | Inference Time (s/20 steps) | 21 | 48 | 129 | 349 |
> | Inference Time (s/50 steps) | 52 | 118 | 323 | 873 |
> | GPU Memory (MB) | 29511 | 32768 | 36663 | 44824 |
> | TFlops/s | 67.9 | 58.2 | 39.4 | 25.4 |
>
> To assess the efficiency of our model with extended token length, we analyze the impact of token length on both efficiency and GPU memory usage. For reference, a 360p 2-second video editing (composed of two videos and one text sequence) uses a token length of 34342 (second column in the table). Experiment results demonstrate that resource consumption growth remains **within a manageable and predictable range**. Specifically, GPU Memory usage exhibits a strong linear relationship with token length. As the sequence length was scaled from 17,652 up to 110,822 tokens, the peak memory footprint increased modestly from 29.5 GB to 44.8 GB. This predictable and relatively slow scaling rate confirms that the extended context itself does not impose an unconstrained memory ceiling. Similarly, inference time shows a systematic, controllable increase with token length.
>
> > **Question 2: 4D RoPE dimension allocation**
> >
>
> We appreciate the reviewer’s careful reading and the question regarding our 4D RoPE allocation. Our design is guided by the token statistics in our training data and by compatibility with the pretrained video backbone.
>
> (1) **Sequential dimension vs. Temporal dimension (12 vs. 4).** In our training corpus, we observe that the length of the sequential token stream (text tokens, control tokens, generate video tokens etc.) is roughly three times the number of temporal tokens (generate video tokens). Consequently, we allocate RoPE dimensions in proportion to this ratio, assigning 12 dimensions to the sequential axis and 4 to the temporal axis.
>
> (2) **Height and Width (56, 56).** For the spatial dimensions, we follow the setting of the pretrained video model, which uses dimension of 56 for both height and width. This choice maintains compatibility with the pretrained representation. When we reduce the spatial RoPE dimensions (e.g., below 56), we observe a noticeable degradation in performance since the model partially “forgetting” the pretrained spatial priors. Thus, we keep 56 for height and width to preserve pretraining information.

---

> ### Author Response · Authors · 2025-11-21
> **Response to Reviewer 3VqN - 2**
>
> > **Question 3 & Weakness 2: Synthetic video editing dataset & Perform on tasks that are not present in its synthetic "teacher" models**
> >
>
> We thank the reviewer for this insightful comment. We would like to clarify that our method is not trained solely on synthetic video editing results from teacher models, nor is its performance bound by a single teacher’s capabilities.
>
> (1) Training data is not limited to synthetic video edits.
>
> While we indeed use a synthetic pipeline (e.g., VACE, DiffuEraser) to obtain part of our video editing data, this subset constitutes only a fraction of the full training corpus. EditVerse is trained jointly on 6.0M image editing samples, 1.9M image generation samples, 3.9M video generation samples, and 288K video editing samples. These additional large-scale image and video generation and editing datasets are independent of the video teacher models and cover distributions that are missing or underrepresented in the synthetic video editing pairs. As a result, the model is exposed to a much broader and richer set of visual transformations than what any single video editing pipeline can provide.
>
> (2) Unified architecture enables cross-modal knowledge transfer for unseen tasks in video editing training data.
>
> Our model is explicitly designed as a unified, interleaved text–image–video framework with full self-attention. This design allows knowledge transfer from the data-rich image domain (both image generation and image editing) to the video domain. In practice, this means that editing skills learned from high-quality image data (e.g., complex material changes, stylization, special effects, multi-step edits) can be applied to videos, **even when such tasks are absent or rare** in the synthetic video editing set. Empirically, we observe that EditVerse can perform video edits that are not directly present in the video editing training data, such as changing materials, adding effects, and composing multiple edits in a single instruction, as illustrated in the teaser and additional qualitative results on our anonymous project page: http://editverse-anonymous.s3-website-us-east-1.amazonaws.com/ . These capabilities indicate that the model is not simply mimicking the teacher pipelines but leveraging its unified representation and joint training to generalize beyond them.
>
> (3) Emergent abilities and performance surpassing teacher-generated “ground truth”.
>
> Section 5.3 analyzes EditVerse's emergent abilities and shows that, for tasks in the video editing training data, EditVerse often produces outputs that **qualitatively surpass the teacher-generated “ground truth” videos** in both visual quality and adherence to instructional guidance. We further support this with ablation studies showing that removing image generation/editing or video generation data significantly degrades video editing performance, confirming that the strong video editing ability arises from our unified design and joint image and video training, rather than being capped by the synthetic video teachers.
>
> To sum up, we observe that EditVerse can (1) produce video edits that are **higher quality than the teacher-generated “ground truth”** and (2) successfully handle **editing tasks that do not exist in the video editing training set**. Therefore, for a concept like “making an object transparent,” even if VACE is poor at this operation, EditVerse can learn the concept from image editing data and apply it to videos, so its video performance on such tasks is not bound by VACE.
>
> > **Question 4: VLM evaluation**
> >
>
> Thank you for your question. For each sampled frame, the VLM jointly rates prompt following, edit quality, and background consistency. Therefore, systematic logical or instruction-misalignment errors that persist across the clip (e.g., consistently placing an object at the wrong spatial location) are typically detected by this metric rather than ignored. For example, the failure case in Fig. 9a receives scores of 1 (prompt following), 3 (edit quality), and 3 (background consistency), indicating that the VLM successfully detects this issue. Moreover, the entire VLM evaluation shows a **Pearson Correlation of 0.84** with the user study, indicating a **very strong positive correlation** between the user study and VLM rankings.
>
> We also agree that rare or highly localized failures may not always be fully captured when only three frames are sampled. For this reason, we (i) complement the VLM score with additional text-alignment and temporal-consistency metrics, and (ii) conduct a large-scale user study whose results are well aligned with the VLM metric. We therefore view the VLM score as a robust, though not perfect, proxy for high-level instruction adherence.

---

> > ### Comment · Reviewer_3VqN · 2025-11-27
> >
> > Thank you for the comprehensive response from the author, I have decided to improve my score.

---

### Official Review · Reviewer_hzXL · 2025-10-29

**Soundness:** 3
**Presentation:** 3
**Contribution:** 3
**Rating:** 6
**Confidence:** 4

**Summary:**

The paper presents EditVerse, a unified video editing model that aims to handle diverse edit types (style change, object manipulation, temporal edits, localized edits, etc.) with a single model. The core modeling idea is to serialize visual content (both images and multi-frame videos) into an interleaved token sequence, and to use a multi-dimensional rotary positional embedding (RoPE) capturing spatial, temporal, and sequential position.

On the evaluation side, the paper introduces EditVerseBench, which focuses on real-world, non-square aspect ratios (horizontal and vertical videos) across multiple editing categories, arguing that prior benchmarks mostly assume square crops and narrow edit types. They also describe a data generation and VLM-based filtering pipeline to curate higher-quality training dataset. Quantitative comparisons are reported using automatic VLM metrics, frame-level visual quality and temporal consistency metrics, plus a human pairwise preference study.

Overall, the paper addresses a practically relevant problem and proposes a promising integration of a unified model design, a new benchmark, and a data curation pipeline. Several points require clarification, in particular the transparency of the evaluation setup, the reporting of the human study, and ablations on key components, but these issues appear to be fixable.

**Strengths:**

The paper addresses a practically relevant problem by aiming to unify diverse instruction-based video editing tasks within a single model. The proposed interleaved visual token representation together with the multi-dimensional RoPE design is technically interesting and potentially reusable beyond this specific application. The introduction of a benchmark with realistic horizontal and vertical video formats and multiple editing categories is valuable for evaluating real-world usage, and the experimental comparisons include both strong instruction-based methods and training-free baselines, which provides a relatively comprehensive view of performance.

**Weaknesses:**

The human evaluation lacks essential reporting details such as annotator composition, annotation protocol, and reliability, making it difficult to assess the credibility of the results. The design choices of the proposed benchmark may be perceived as favoring the method, and the paper does not sufficiently analyze scenarios in which simpler, training-free baselines still outperform the model. In addition, the contribution of the positional encoding components is not well-isolated through ablations, and the automatic evaluation pipeline, including data filtering and VLM-based scoring prompts, is not fully transparent, which limits the reproducibility of the results.

**Questions:**

1. **Human evaluation protocol**

   More details are needed on the human preference study: how many annotators were involved, their backgrounds, how annotation was distributed per person, and whether any quality control or inter-annotator agreement was assessed. Such information would increase the credibility of the human study results.

2. **Clarification on the Qwen2-VL–based data filtering process**
   The description of the VLM filtering process remains vague. Could you specify the exact score threshold used to retain samples, provide an overview of score distributions before and after filtering (ideally across data sources), and comment on whether Qwen2-VL frequently misjudged quality (e.g., bad edits retained or good edits discarded), even approximately? Greater transparency here would strengthen the argument for the data pipeline.

3. **Human rating in the main results table**
   For the core quantitative comparison (e.g., Table 2), it would be valuable to include a human rating or user preference column alongside the automatic metrics, similar to prior works such as TokenFlow, Tune-A-Video, and FateZero. This would provide a more direct comparison from a human perspective and make the main results more informative, rather than relying solely on automatic metrics.

4. **Benchmark construction and possible bias**

   Since EditVerseBench excludes square videos entirely, it would be helpful to clarify the rationale for not including them alongside horizontal and vertical formats. This concern is reinforced by the fact that in Table 6, EditVerse does not always outperform training-free baselines on V2VBench, where some simpler methods achieve competitive or better results. It would be useful to discuss in which scenarios such lightweight baselines remain preferable in practice, and how the proposed benchmark design avoids unintentionally favoring EditVerse.

5. **RoPE ablations**
   As the multi-dimensional RoPE is positioned as a key architectural contribution, more direct evidence isolating its effect would be useful. Even a small-scale study comparing variants that remove temporal RoPE, remove spatial RoPE, or use only standard sequential RoPE would help substantiate the importance of each component.

6. **Data filtering transparency**
    The data pipeline uses a VLM to assign 0–10 quality scores (instruction adherence, temporal consistency, artifact severity, etc.) and then thresholds these scores to filter large editing datasets.

   - Please show the score distribution before vs. after filtering.
   - How was the threshold chosen, and how sensitive is final model performance to that threshold?
   - Do you have an estimate of “false positives” (bad edits that passed) and “false negatives” (good edits that were filtered out)?
      Since the dataset is part of the claimed contribution, a brief audit would strengthen the narrative.

7. **Reproducibility of automatic evaluation**
   To facilitate reproducibility, would you consider releasing the exact prompt templates and inference configurations used for the VLM-based automatic evaluation? Without these details, the community may find it difficult to replicate the automatic scores.

8. **Minor fixes for clarity**

   - The training loss formula should be corrected to:
     $$
     \mathcal{L} = \mathbb{E}_{t,X_0,X_1} \left\| u_\Theta(X_t, t) - (X_1 - X_0) \right\|^2.
     $$

   - The naming and ordering of the RoPE components in the text (around lines 200–221) do not match Figure 2. The text first describes “Height and Width Dimensions” before sequential and temporal, while Figure 2 uses the order spatial → temporal → sequential. Please unify terminology and ordering to avoid confusion.

   - In Table 9, the symbol used to mark the method is “†”, but the note below the table refers to “‡ uses LLM-rewritten prompts.” The marker in the table and the footnote symbol should be consistent.

   - The citation for “TokenFlow” is incorrect. The paper currently cites *“Tokenflow: Unified Image Tokenizer for Multimodal Understanding and Generation”*, which is unrelated to video editing. It should cite **“TokenFlow: Consistent Diffusion Features for Consistent Video Editing”**. The incorrect reference appears in line 348–349, Table 3, and line 818.

---

> ### Author Response · Authors · 2025-11-21
> **Response to Reviewer hzXL - 1**
>
> We sincerely thank the reviewer for the encouraging and insightful feedback, and for recognizing our problem setting, the technical contribution of our interleaved visual token representation and multi-dimensional RoPE design, as well as the value of our benchmark and comprehensive comparisons. Below, we address each concern in detail and clarify the changes we made in the revised version (marked in red).
>
> > **Question 1 & Weakness 1: Human evaluation protocol**
> >
>
> We appreciate the reviewer’s suggestion and have added more details about our human preference study in the revised manuscript (marked in red). The user study was outsourced to a professional external vendor, who recruited 20 annotators coming from diverse non-expert backgrounds. Each comparison pair was independently evaluated by 3 different annotators, and each annotator labeled 150 comparison pairs in total. To faithfully capture end-user preferences, we intentionally kept the instructions minimal and user-centric, providing only the brief prompt shown in Appendix A.1, without additional technical guidance. This setup was chosen to reflect how typical users would judge the outputs rather than imposing task-specific expertise. Although we did not compute a formal inter-annotator agreement statistic, the redundancy of three independent judgments per pair helps mitigate noise and increases the robustness of the aggregated preferences. We will also clarify in the paper that concrete visualization examples of the annotation interface are available at [htt](http://editverse-anonymous.s3-website-us-east-1.amazonaws.com/)p://editverse-anonymous.s3-website-us-east-1.amazonaws.com/, and that the raw annotation results for RunwayAleph are available in Evaluation/UserStudy.xlsx, which together further support the credibility and transparency of our human study.
>
> > **Question 2 & Weakness 3 & Question 6: Clarification on the Qwen2-VL–based data filtering process & Data filtering transparency**
> >
>
> We thank the reviewer for pointing out this ambiguity and apologize for the lack of detail in the current draft. We used a strong off-the-shelf vision–language model (Qwen2.5-VL) for data filtering. Concretely, for each generated sample, we asked the VLM to assign a score in [0,10] along eight dimensions (the prompt is reported in Sec. A.1): instruction adherence, context preservation, sharpness, temporal consistency, artifacts, object integrity, aesthetics, and physical plausibility. Higher scores indicate better quality. To assess the reliability of these scores, we bucketed generated samples by VLM score and conducted manual inspection for each dimension. Samples in high-score buckets consistently exhibited faithful edits, whereas low-score buckets contained frequent instruction failures, temporal flickering, and visual artifacts. Based on this analysis, we chose per-dimension thresholds that align with human judgments of acceptable quality while retaining enough data for training. Specifically, a sample is kept only if it satisfies 5 among 8 threshold criteria: instruction adherence > 6, context preservation > 5, sharpness > 7, temporal consistency > 6, artifacts > 6, object integrity > 7, aesthetics > 4, and physical plausibility > 5. These thresholds were selected by jointly examining empirical score distributions and visually inspecting borderline cases near candidate cutoffs. The average score across the whole dataset increased from 3.88 to 5.63 after filtering, and the average false-positive and false-negative rates are around 20%. Upon manual verification of 100 samples from our data generation pipeline after the VLM filtering, we find that **67%/19%/14% fall into good/medium/bad**, where “good” signifies that the editing is correct with minimal artifacts, “medium” signifies that the editing is correct but with some artifacts or effects (like shadows or reflections) left over, and “bad” means that the editing is incorrect or contain notable artifacts. 86% of the samples are considered usable after manual inspection (classified as good or medium), providing sufficient learning signals for the model to understand video editing tasks. Note that high-quality image generation, image editing, and video generation supplement these synthetic data and significantly improve final performance.
>
> > **Question 3: Human rating in the main results table**
> >
>
> We thank the reviewer for this helpful suggestion. In the revised version of the paper, we have updated the main quantitative comparison tables (Table 2) to include an additional column reporting human ratings scores alongside the automatic metrics (marked in red).

---

> ### Author Response · Authors · 2025-11-21
> **Response to Reviewer hzXL - 2**
>
> > **Question 4 & Weakness 2: Benchmark construction and possible bias**
> >
>
> We thank the reviewer for raising this point. We did not include square videos in EditVerseBench for two main reasons. (1) Square videos are already extensively covered in existing benchmarks (e.g., V2VBench and TGVE), which can be directly used for evaluation, and we also provide their evaluation results in Table 3 and Table 5. Our goal was to provide a complementary benchmark focusing on previously under-explored cases. (2) In practical applications, square videos are much less common than vertical (e.g., mobile/social) and horizontal (e.g., cinematic) formats, which dominate real-world usage. We therefore prioritized horizontal and vertical layouts to better match typical deployment scenarios.
>
> Regarding Table 6 and the V2VBench results: (1) EditVerse is not trained on square videos, so the results reported there should be viewed as a zero-shot transfer to this format rather than an in-domain evaluation. (2) The V2VBench automatic metrics do not fully align with human perception of editing quality. While they capture aspects such as temporal consistency and frame-level quality, they often miss higher-level semantic and perceptual factors. As shown in our visual comparisons of EditVerseBench on the anonymous webpage http://editverse-anonymous.s3-website-us-east-1.amazonaws.com/ , some training-free methods such as TokenFlow can achieve competitive scores on specific metrics, yet their visual results are noticeably worse than EditVerse in terms of edit fidelity and overall coherence.
>
> We agree that lightweight training-free baselines can still be preferable in practice when only small local edits are required. Our intention with EditVerseBench is not to favor EditVerse, but to stress more challenging, realistic editing scenarios on horizontal and vertical videos under a unified protocol. All methods, including training-free baselines, are evaluated under the same data, prompts, and metrics, and we additionally provide qualitative results to expose cases where automatic scores and human judgments diverge.
>
> > **Question 5 & Weakness 3: RoPE ablations**
> >
>
> We thank the reviewer for this insightful suggestion and agree that better isolating the effect of the multi-dimensional RoPE is important. In the current submission, the reported results in the ablation study are obtained with temporal and spatial RoPE enabled and without sequential RoPE. We want to highlight that both sequential and spatial RoPEs are required to inform essential positional information, which is also used in previous video generation models (e.g., Hunyuan Video[0]). Removing either one leads to failed video generation in our preliminary tests (e.g., generating pure noise video without content), as the model effectively loses the ability to disambiguate tokens across time or space. Thus, we only focus on removing sequential RoPE in our experiments (as shown in Table 5). Due to the limited time available during the rebuttal period, we were unable to run a systematic ablation with the two requested variants (no temporal RoPE, no spatial RoPE). However, we appreciate the feedback for a more thorough analysis.
>
> [0] Kong W, Tian Q, Zhang Z, et al. Hunyuanvideo: A systematic framework for large video generative models[J]. arXiv preprint arXiv:2412.03603, 2024.
>
> > **Question 7 & Weakness 3: Reproducibility of automatic evaluation**
> >
>
> We thank the reviewer for emphasizing the importance of reproducibility. We will **open-source the corresponding evaluation code**, which will contain the complete set of prompts, hyperparameters, and execution scripts. We hope this will make it straightforward for the community to reproduce our automatic scores and adapt the evaluation pipeline to future work.
>
> > **Question 8: Minor fixes for clarity**
> >
>
> We thank the reviewer for carefully pointing out these issues. We have corrected all the issues in the revised version. All the changes are highlighted in red in the revised manuscript.

---

> > ### Comment · Reviewer_hzXL · 2025-11-25
> >
> > Thanks for the author's thoughtful response. I decide to maintain my positive evaluation of this work.

---

### Official Review · Reviewer_ijpS · 2025-10-30

**Soundness:** 2
**Presentation:** 3
**Contribution:** 2
**Rating:** 4
**Confidence:** 3

**Summary:**

This paper proposes EditVerse, a unified framework for image and video generation and editing. The core idea is to represent text, image, and video as a single interleaved token sequence, processed by a full self-attention transformer. A 4D ROPE position embedding is proposed to encode spatial, temporal, and sequence positions. To address video editing data scarcity, the authors build a synthetic video editing pipeline to collect 0.2 million video editing data for model training. Experiments shows that their model can suprpass existing open-source and commercial models across various tasks.

**Strengths:**

1. The paper is well-written, and the proposed framework is clearly presented and easy to understand.

2. The visualized qualitative results are impressive and effectively demonstrate the model’s capabilities.

**Weaknesses:**

1. The main weakness of this work lies in the lack of substantial technical novelty. The proposed pipeline appears highly similar to recent multimodal generative frameworks (e.g., Bagel, OmniGen). The novelty mainly resides in engineering integration rather than in proposing a fundamentally new modeling formulation.

2. In my opinion, the key contribution of the paper is to show that a unified video editing/generation framework can be implemented. However, the authors do not commit to releasing the engineering details, dataset collection pipeline, or the collected dataset itself. Without open sourcing or releasing enough details for reproduction, their engineering contribution becomes less impactful.

3. The justification for the necessity of the proposed interleaved representation remains insufficient. Although the authors include some ablations, it is still unclear why interleaving tokens is fundamentally better than other existing designs (e.g., cross-attention conditioning, multi-encoder structures).

**Questions:**

Refer to weakness.

---

> ### Author Response · Authors · 2025-11-21
> **Response to Reviewer ijpS - 1**
>
> We sincerely thank the reviewer for the thoughtful and constructive feedback, as well as for the positive assessment of our comprehensive experiments and presentation. Below, we address each concern in detail and clarify the changes we made in the revised version (marked in red).
>
> > **Weakness 1: Technical novelty vs. engineering integration**
> >
>
> We thank the reviewer for the comment and agree that EditVerse is built upon transformer-based multimodal modeling. However, our contribution extends beyond engineering and scaling, with conceptual advances and technical novelty as follows:
>
> (1) From the perspective of video editing models, EditVerse uses a **fully unified in-context formulation, which has not been systematically explored** **for video generation previously**, let alone in **instruction-based video editing**. Specifically, EditVerse treats text, images, and videos as a single interleaved token sequence processed by full self-attention, without any task-specific heads or modality-specific decoders. In contrast, prior instruction-based video editing methods such as UNIC[1], EVE[2], and InsV2V[3] all adopt cross-attention layers to input text conditions.
>
> (2) From the broader multimodal modeling perspective, most unified generative frameworks either focus only on image generation and editing (e.g., Bagel[4], Transfusion[5]), or discuss video generation but not video editing and **do not analyze how large-scale image editing data can systematically benefit video editing** (e.g., Show-o2[6], Emu3[7]). EditVerse jointly handles image and video generation and editing within a single model and explicitly examines how abundant image editing data can compensate for scarce video editing data.
>
> (3) Architecturally, we introduce a **four-dimensional Rotary Positional Embedding** that jointly indexes sequential, temporal, height, and width dimensions within the same transformer. This design is explicitly constructed for interleaved text–image–video tokens and is crucial for enabling the model to distinguish and relate tokens across modalities, spatial locations, and time.
>
> (4) Finally, we go beyond showing that the unified model “works”: **we analyze why it works**. We demonstrate that EditVerse exhibits emergent abilities, such as performing editing tasks not explicitly present in the video editing training set (e.g., complex material or weather changes, multi-step edits), and in some cases generating edits that surpass the quality of the pseudo-ground-truth training pairs. Our data ablations further reveal that both image generation/editing data and video generation data are indispensable, removing either significantly degrades video editing performance, especially in instruction alignment and temporal consistency. These findings suggest that the unified sequence design leads to qualitatively new behaviors rather than a straightforward scaled-up implementation of existing paradigms.
>
> [1] Ye Z, He X, Liu Q, et al. UNIC: Unified In-Context Video Editing[J]. arXiv preprint arXiv:2506.04216, 2025.
>
> [2] Singer U, Zohar A, Kirstain Y, et al. Video editing via factorized diffusion distillation[C]//European Conference on Computer Vision. Cham: Springer Nature Switzerland, 2024: 450-466.
>
> [3] Cheng J, Xiao T, He T. Consistent video-to-video transfer using synthetic dataset[J]. arXiv preprint arXiv:2311.00213, 2023.
>
> [4] Deng C, Zhu D, Li K, et al. Emerging properties in unified multimodal pretraining[J]. arXiv preprint arXiv:2505.14683, 2025.
>
> [5] Zhou C, Yu L, Babu A, et al. Transfusion: Predict the next token and diffuse images with one multi-modal model[J]. arXiv preprint arXiv:2408.11039, 2024.
>
> [6] Xie J, Yang Z, Shou M Z. Show-o2: Improved Native Unified Multimodal Models[J]. arXiv preprint arXiv:2506.15564, 2025.
>
> [7] Wang X, Zhang X, Luo Z, et al. Emu3: Next-token prediction is all you need[J]. arXiv preprint arXiv:2409.18869, 2024.
>
> > **Weakness 2: Detailed release**
> >
>
> We appreciate the reviewer’s recognition of our unified video editing/generation framework and the importance of reproducibility. We are currently re-implementing our data pipeline with the intention of releasing it publicly. In addition, we will release EditVerseBench together with our complete evaluation protocol (including task definitions, input/output formats, prompts, and evaluation scripts), so that future works can directly compare under the same setting.
>
> Moreover, in the current version of the paper, we already provide architectural and training details in both the main text and the appendix (e.g., model components, data sources, and training configurations) to facilitate independent reproduction of our results before code release. In the revised version, we will further refine and expand these descriptions, adding more implementation notes and ablation details, to make reproduction and fair comparison as straightforward as possible.

---

> ### Author Response · Authors · 2025-11-21
> **Response to Reviewer ijpS - 2**
>
> > **Weakness 3: Interleaved representation**
> >
>
> We thank the reviewer for raising this important point. We admit that, in relatively simple single-video editing scenarios, the gap between interleaved designs and cross-attention or multi-encoder designs may be smaller. But we would like to clarify why we believe the interleaved representation is necessary for our setting:
>
> (1) EditVerse is designed to handle **a large family of tasks within a single model**: text-to-image generation, text-to-video generation, instruction-based image and video editing, control-to-video (pose / depth / sketch), propagation, inpainting, etc., all with images and videos of arbitrary resolution and duration. In our formulation, every input (instruction text, source images/videos, masks, reference frames) and every output are mapped to the same interleaved token sequence and processed by a single backbone, without task-specific branches or encoders. This is what allows a single model to operate consistently across the 20+ editing tasks and various generation tasks we evaluate. In contrast, cross-attention conditioning or multi-encoder structures typically assume a fixed set of “main” and “conditioning” streams and require architectural changes or extra heads when adding new input types (e.g., additional reference videos, masks, or control signals), which makes it difficult to scale to a unified, in-context setting.
>
> (2) Our interleaved sequence **preserves the natural ordering of text, image, and video**, where the text describing an edit is usually placed near the corresponding visual content. Together with the 4D RoPE over sequential, temporal, height, and width axes, full self-attention explicitly exploits local text–vision co-occurrence. This leads to better text alignment and edit quality than treating text as a separate conditioning stream. Empirically, Table 5 shows that removing either (i) the interleaved input (placing all vision tokens at the end) or (ii) the sequential RoPE significantly degrades VLM editing quality and CLIP-based text alignment, even though the model capacity and training data are identical. This indicates that the interleaved representation, not just the amount of data, is crucial.
>
> (3) Our design is intentionally forward-looking: large-scale web corpora naturally contain mixed streams of text, images, and videos in arbitrary order. Representing them as a single interleaved sequence aligns the model’s architecture with this native data format and makes it straightforward to extend EditVerse to longer, more complex multimodal contexts (e.g., multiple demonstrations plus a new query video) without changing the backbone.

---

> > ### Comment · Reviewer_ijpS · 2025-11-22
> > **Response to authors**
> >
> > Thanks for the authors’ response. I still have some doubts about why interleaved representations are better than alternative designs, especially given that high-quality video or image-editing data is both scarce and expensive, and video editing itself is highly complex. Nevertheless, I appreciate the authors’ effort in clarifying the data structure and model architecture in detail.
> >
> > Although I still hold some reservations, I would like to raise my score to 6 to encourage the authors to release their model or data pipeline, ideally by providing an anonymous code link before the end of rebuttal.
> >
> > Gool luck, hope to see this work in ICLR conference.

---

### Official Review · Reviewer_U1BY · 2025-11-03

**Soundness:** 3
**Presentation:** 3
**Contribution:** 3
**Rating:** 6
**Confidence:** 4

**Summary:**

This paper introduces EditVerse, a unified multimodal framework for text-guided image and video generation and editing. The model represents text, image, and video modalities as a single interleaved token sequence and applies self-attention for in-context learning and cross-modal reasoning. To address the lack of video editing data, the authors design a large-scale automated data generation pipeline. Extensive experiments demonstrate that EditVerse achieves strong performance and exhibits emergent editing capabilities that generalize beyond the training set.

**Strengths:**

* Comprehensive experiments with solid quantitative and qualitative evaluations.
* Clear presentation and well-structured writing.
* The unified token representation for text, image, and video is conceptually elegant.
* Interesting empirical insight: the model demonstrates emergent abilities on unseen editing tasks, suggesting strong cross-modal generalization.

**Weaknesses:**

* While the unified sequence formulation is elegant, it primarily extends existing transformer-based multimodal modeling paradigms. The contribution lies more in engineering integration and data scaling than in introducing fundamentally new learning principles.
* The model requires ~30 GB GPU memory and ~118 seconds to edit a single 360p video on an A100 80 GB GPU. This raises serious scalability concerns for long-duration or high-resolution videos, as well as practical deployment constraints.
* The approach concatenates all modality tokens into one long sequence, resulting in $O(L^2)$ attention cost. It is unclear how EditVerse maintains efficiency with long inputs, multiple video clips, or multi-turn text–video interleaving. No efficiency analysis (FLOPs, latency, or memory growth curves) is provided.
* The use of single-layer linear projectors for modality alignment may be oversimplified. There is no ablation on alternative projector depths or modality-specific encoders to validate whether this minimal design is sufficient for cross-modal alignment.
* The video editing dataset is entirely generated via an automated pipeline using pretrained models (Grounded-SAM-2, VACE, ReCamMaster, etc.). This synthetic data may not reflect the diversity and imperfections of real-world edits, leading to overfitting on artificial editing patterns.
* Each data generation stage depends on prior model outputs, so artifacts such as inaccurate masks, poor inpainting, or unrealistic motion can cascade. The paper provides no quantitative analysis or quality control to assess data noise accumulation.
* The paper mentions using a VLM to assign quality scores for filtering generated data, but it is unclear whether the VLM was adapted for the data filtering, how accurate its scores are, or how thresholds were chosen.

**Questions:**

* How is the VLM used for data scoring and filtering? Was it used off-the-shelf or fine-tuned for editing relevance? What is its reliability or correlation with human judgment?
* How does model performance and runtime scale with sequence length (e.g., multi-video input or long textual instructions)? Can efficiency be improved beyond full-sequence attention?
* Have you compared the single-layer modality projectors against deeper or non-linear alternatives?
* The reported improvements are modest; how sensitive are the evaluation metrics, and do the differences translate into perceptual gains?
* Given that the data pipeline heavily depends on pretrained vision models, how do you ensure the resulting dataset and EditVerse itself do not inherit or amplify their biases?

---

> ### Author Response · Authors · 2025-11-21
> **Response to Reviewer U1BY - 1**
>
> We sincerely thank the reviewer for the thoughtful and constructive feedback, as well as for the positive assessment of our comprehensive experiments, presentation, and the conceptual elegance and emergent capabilities of our unified multimodal formulation. Below, we address each concern in detail and clarify the changes we made in the revised version (marked in red).
>
> > **Weakness 1: Conceptual novelty vs. engineering integration and data scaling**
> >
>
> We thank the reviewer for the comment and agree that EditVerse is built upon transformer-based multimodal modeling. However, our contribution extends beyond engineering and scaling, with conceptual advances and technical novelty as follows:
>
> (1) From the perspective of video editing models, EditVerse uses a **fully unified in-context formulation, which has not been systematically explored** **for video generation previously**, let alone in **instruction-based video editing**. Specifically, EditVerse treats text, images, and videos as a single interleaved token sequence processed by full self-attention, without any task-specific heads or modality-specific decoders. In contrast, prior instruction-based video editing methods such as UNIC[1], EVE[2], and InsV2V[3] all adopt cross-attention layers to input text conditions.
>
> (2) From the broader multimodal modeling perspective, most unified generative frameworks either focus only on image generation and editing (e.g., Bagel[4], Transfusion[5]), or discuss video generation but not video editing and **do not analyze how large-scale image editing data can systematically benefit video editing** (e.g., Show-o2[6], Emu3[7]). EditVerse jointly handles image and video generation and editing within a single model and explicitly examines how abundant image editing data can compensate for scarce video editing data.
>
> (3) Architecturally, we introduce a **four-dimensional Rotary Positional Embedding** that jointly indexes sequential, temporal, height, and width dimensions within the same transformer. This design is explicitly constructed for interleaved text–image–video tokens and is crucial for enabling the model to distinguish and relate tokens across modalities, spatial locations, and time.
>
> (4) Finally, we go beyond showing that the unified model “works”: **we analyze why it works**. We demonstrate that EditVerse exhibits emergent abilities, such as performing editing tasks not explicitly present in the video editing training set (e.g., complex material or weather changes, multi-step edits), and in some cases generating edits that surpass the quality of the pseudo-ground-truth training pairs. Our data ablations further reveal that both image generation/editing data and video generation data are indispensable, removing either significantly degrades video editing performance, especially in instruction alignment and temporal consistency. These findings suggest that the unified sequence design leads to qualitatively new behaviors rather than a straightforward scaled-up implementation of existing paradigms.
>
> [1] Ye Z, He X, Liu Q, et al. UNIC: Unified In-Context Video Editing[J]. arXiv preprint arXiv:2506.04216, 2025.
>
> [2] Singer U, Zohar A, Kirstain Y, et al. Video editing via factorized diffusion distillation[C]//European Conference on Computer Vision. Cham: Springer Nature Switzerland, 2024: 450-466.
>
> [3] Cheng J, Xiao T, He T. Consistent video-to-video transfer using synthetic dataset[J]. arXiv preprint arXiv:2311.00213, 2023.
>
> [4] Deng C, Zhu D, Li K, et al. Emerging properties in unified multimodal pretraining[J]. arXiv preprint arXiv:2505.14683, 2025.
>
> [5] Zhou C, Yu L, Babu A, et al. Transfusion: Predict the next token and diffuse images with one multi-modal model[J]. arXiv preprint arXiv:2408.11039, 2024.
>
> [6] Xie J, Yang Z, Shou M Z. Show-o2: Improved Native Unified Multimodal Models[J]. arXiv preprint arXiv:2506.15564, 2025.
>
> [7] Wang X, Zhang X, Luo Z, et al. Emu3: Next-token prediction is all you need[J]. arXiv preprint arXiv:2409.18869, 2024.

---

> ### Author Response · Authors · 2025-11-21
> **Response to Reviewer U1BY - 2**
>
> > **Weakness 2 & Question 2: Efficiency and scalability**
> >
>
> We agree with the reviewer that efficiency and scalability are essential, as we noted in the limitations section. However, our work was not explicitly designed to improve efficiency, and the reported ~30GB of memory usage and ~118 seconds of inference time reflect the model's raw, unoptimized performance using 50 steps. There are several practical ways to improve it, which we plan to explore in future work:
>
> (1) **Using a higher-compression VAE.** In our model, we use a VAE with relatively low compression (8× spatial downsampling), resulting in a large number of visual tokens. Recent VAEs can achieve 16× spatial compression. If we replace our VAE with 16× spatial compression rate models, the token length can be reduced to about one quarter, thereby directly lowering the attention cost.
>
> (2) **Dynamic token selection.** We can introduce a dynamic token selection mechanism that adaptively keeps only the most important context tokens and prunes redundant ones. This can reduce the effective sequence length for full attention, as also explored in recent works such as FullDiT2[8].
>
> (3) **Distillation for faster inference.** We can further apply model distillation and step distillation to reduce both the number of diffusion steps and the model size, which can speed up generation. (e.g., Self Forcing[9])
>
> (4) **Efficient attention mechanisms.** Future work could explore more efficient attention mechanisms (e.g., linear attention, Mamba attention) to reduce the computational burden without compromising the model's cross-modal learning capabilities.
>
> We have expanded the discussion of these directions in the revised version (Section A.4) to clarify that there are clear paths to make EditVerse more practical in real-world settings.
>
> [8] He X, Liu Q, Ye Z, et al. Fulldit2: Efficient in-context conditioning for video diffusion transformers[J]. arXiv preprint arXiv:2506.04213, 2025.
>
> [9] Huang X, Li Z, He G, et al. Self Forcing: Bridging the Train-Test Gap in Autoregressive Video Diffusion[J]. arXiv preprint arXiv:2506.08009, 2025.
>
> > **Weakness 3 & Question 2: Efficiency analysis**
> >
>
> We thank the reviewer for pointing out the absence of an efficiency analysis. We incorporate a more thorough examination of efficiency, along with relevant discussions and results, in the revised version of Section A.4 (marked in red). We summarize the analysis here:
>
> | Token Length | 17652 | 34342 | 63662 | 110822 |
> | --- | --- | --- | --- | --- |
> | Inference Time (s/20 steps) | 21 | 48 | 129 | 349 |
> | Inference Time (s/50 steps) | 52 | 118 | 323 | 873 |
> | GPU Memory (MB) | 29511 | 32768 | 36663 | 44824 |
> | TFlops/s | 67.9 | 58.2 | 39.4 | 25.4 |
>
> To assess the efficiency of our model with extended token length, we analyze the impact of token length on both efficiency and GPU memory usage. For reference, a 360p 2-second video editing (composed of two videos and one text sequence) uses a token length of 34342 (second column in the table). Experiment results demonstrate that resource consumption growth remains **within a manageable and predictable range**. Specifically, GPU Memory usage exhibits a strong linear relationship with token length. As the sequence length was scaled from 17,652 up to 110,822 tokens, the peak memory footprint increased modestly from 29.5 GB to 44.8 GB. This predictable and relatively slow scaling rate confirms that the extended context itself does not impose an unconstrained memory ceiling. Similarly, inference time shows a systematic, controllable increase with token length.
>
> > **Weakness 4 & Question 3: Lack of ablation for single-layer modality projectors**
> >
>
> We appreciate this suggestion and agree that the design of the modality projectors is important. Due to the limited time available during the rebuttal period, we are unfortunately unable to retrain the model and run a full ablation with deeper projector networks. However, the current results already show that a **single-layer MLP for projection is sufficient to achieve state-of-the-art performance**, suggesting that this simple design is effective in practice. We agree that more profound or more complex projectors may further improve performance, and we will systematically study projector depth and architecture in future work.

---

> ### Author Response · Authors · 2025-11-21
> **Response to Reviewer U1BY - 3**
>
> > **Weakness 5 & Question 5: Synthetic video editing dataset distribution bias**
> >
>
> We want to emphasize that our training data is not limited to synthetic video editing; we also use **large amounts of real video generation data, as well as image generation and image editing data**. These additional data sources help fill distribution gaps in the synthetic video editing set. As shown in our ablation studies, real video generation data and image generation/editing data contribute significantly to the final performance. This is also a key contribution of our work: EditVerse can effectively leverage readily available image and video generation/editing data to compensate for the scarcity of real video editing annotations.
>
> > **Weakness 6 & Question 5: Synthetic video editing dataset noise accumulation**
> >
>
> We thank the reviewer for pointing out the potential issue of error propagation in multi-stage synthetic data generation. We agree that inaccuracies in intermediate outputs (e.g., masks or inpainting results) could accumulate and degrade data quality, and that real video editing data would be more ideal. However, collecting sufficient high-quality real video editing pairs to train a large foundation model is extremely challenging in practice due to annotation costs and time, so we rely on a synthetic pipeline combined with strict quality control. For the **multi-step video editing pipelines,** such as Object Removal and Addition and Object Replacement, the stages that later edits depend on are purposely **built on top of high-precision detection and segmentation models**. In addition, we set relatively high confidence thresholds to further reduce the chance that low-quality masks enter the training set. To quantitatively assess the impact of these choices, we randomly sampled 200 synthetic video editing pairs from our training data and manually verified the correctness of the detected mask. We observe a 95.5% success rate, indicating the first annotation step’s error is small.
>
> Moreover, it is worth noting that the video editing data has been filtered, and it **represents only a small fraction of our overall training data** (288K video editing pairs out of many millions of total image/video generation and editing samples). As a result, the model’s behavior is strongly regularized by large amounts of high-quality image generation/editing and video generation data, which further mitigates the impact of residual noise in the synthetic video editing subset.
>
> > **Weakness 7 & Question 1: VLM-based filtering**
> >
>
> We thank the reviewer for pointing out this ambiguity and apologize for the lack of detail in the current draft. Our data-filtering VLM was not fine-tuned. We used a strong off-the-shelf vision–language model (Qwen2.5-VL). Concretely, for each generated sample, we asked the VLM (with the prompt template shown in Section A.1) to assign a score in [0,10] along eight dimensions: instruction adherence, context preservation, sharpness, temporal consistency, artifacts, object integrity, aesthetics, and physical plausibility. Higher scores indicate better quality. To assess the reliability of these scores, we bucketed generated samples by VLM score and conducted manual inspection for each dimension. Samples in high-score buckets consistently exhibited faithful edits, whereas low-score buckets contained frequent instruction failures, temporal flickering, and visual artifacts. Based on this analysis, we chose per-dimension thresholds that align with human judgments of acceptable quality while retaining enough data for training. Specifically, a sample is kept only if it satisfies 5 among 8 threshold criteria: instruction adherence > 6, context preservation > 5, sharpness > 7, temporal consistency > 6, artifacts > 6, object integrity > 7, aesthetics > 4, and physical plausibility > 5. These thresholds were selected by jointly examining empirical score distributions and visually inspecting borderline cases near candidate cutoffs. The average score across the whole dataset increased from 3.88 to 5.63 after filtering, and the average false-positive and false-negative rates are around 20%. Upon manual verification of 100 samples from our data generation pipeline after the VLM filtering, we find that **67%/19%/14% fall into good/medium/bad**, where “good” signifies that the editing is correct with minimal artifacts, “medium” signifies that the editing is correct but with some artifacts or effects (like shadows or reflections) left over, and “bad” means that the editing is incorrect or contain notable artifacts. 86% of the samples are considered usable after manual inspection (classified as good or medium), providing sufficient learning signals for the model to understand video editing tasks. Note that high-quality image generation, image editing, and video generation supplement these synthetic data and significantly improve final performance.

---

> ### Author Response · Authors · 2025-11-21
> **Response to Reviewer U1BY - 4**
>
> > **Question 4: Evaluation & perceptual gains**
> >
>
> The observed automatic evaluation gains consistently translate into perceptual improvements, as verified by both a user study and qualitative visualizations (in the supplementary material). Among these, the VLM-based evaluation is most aligned with human perception, the ranking of different methods given by the VLM closely matches the ranking obtained from human raters in our user study, indicating that the measured improvements are not merely numerical noise but correspond to changes that users can actually see (as shown in Table 1). Moreover, the entire VLM evaluation shows a **Pearson Correlation of 0.84** with the user study, indicating a **very strong positive correlation** between the user study and VLM rankings.
>
> | Model | EditVerse | Runway Aleph | Lucy Edit Dev | InsV2V | TokenFlow | STDF |
> | --- | --- | --- | --- | --- | --- | --- |
> | User Study (rank) | 1 | 2 | 3 | 4 | 5 | 6 |
> | VLM Evaluation (rank) | 1 | 2 | 3 | 5 | 4 | 6 |
>
> *Table: The rank of each model in the user study and VLM evaluation*
>
> At the same time, the other automatic metrics are designed to capture specific aspects of generation quality, and the improvements we report are consistent across these dimensions. For example, gains on temporal-consistency-related scores correspond to visibly fewer flickering and unstable regions in the video, and higher sharpness and artifact scores align with more precise object boundaries and fewer distortions in edited areas. These trends are clearly observable in the visualizations we provide.

---

> > ### Comment · Reviewer_U1BY · 2025-11-27
> >
> > I appreciate the authors’ efforts in addressing my concerns regarding the model design, data curation and evaluation. I still have some reservations about the computational complexity for practical use, but overall I am positive about the paper’s contributions and lean toward accepting it for ICLR 2026.

---

### Meta-Review · Area_Chair_8Bwm · 2025-12-24

**Summary:**

The paper proposes a novel video-to-video editing framework. The framework is feedforward. The framework is able to take text, image, and video tokens as input without restrictions on the order. The paper makes a contribution to a topic of wide interest and importance. While there was an obvious trend to work towards that direction, the submitted paper demonstrated the first credible realization of such an architecture.

On the negative side, the following aspects limit the contribution:
1) The benchmark possibly or even likely consists of editing examples that favor Editverse. This is not a significant problem for demonstrating the improvement over the state of the art, but it makes the benchmark questionable for future comparisons against Editverse. The authors are highly encouraged to release at least inference code so future comparisons with EditVerse are possible. One reason for the possibility of cherry-picking is that the benchmark only contains examples where EditVerse does well. The base video model is fairly small and the error rate of the base model alone should lead to several failed edits. It would be good to see additional information in the final version.

2) The paper does not sufficiently describe the details necessary to reproduce the work. This is not specific to this paper, but rather to all similar papers that describe a larger system and contain several tricks to make things work, but don't release code and data. Even the pre-training alone is highly non-trivial and requires a lot of compute, data curation, and engineering to make it work. This could be a separate paper or technical report that is summed up in 1-2 sentences. The limited release of information limits the impact the work will have.

**Reviewer Concerns:**

The reviewer also listed additional concerns that were partially addressed in the rebuttal. One of them was the high computation cost. I think the concern is valid, but it should not influence the evaluation of this paper.
The two concerns above remain unresolved after the rebuttal. The first concern does not significantly impact the paper's contribution.

The second concern is valid, and the limited release of information is problematic. I would argue that it is acceptable because it is permitted according to conference policy.

Overall, some reviewers undervalue the paper's importance.

**Reviewer Scores:**

The reviewer scores are positive. The average score after the rebuttal is predicted to be slightly above 6 based on the given set of reviewers.

---

### Decision · Program_Chairs · 2026-01-26

Accept (Oral)